

# Assessment of cloud properties from the reanalysis with satellite observations over East Asia

Bin Yao[1,2], Chao Liu[1,2], Yan Yin[1,2], Zhiquan Liu[3], Chunxiang Shi[4], Hironobu Iwabuchi[5], and Fuzhong Weng[6]

[1]Collaborative Innovation Center on Forecast and Evaluation of Meteorological Disasters, Nanjing University of Information Science & Technology, Nanjing 210044, China

[2]Key Laboratory for Aerosol-Cloud-Precipitation of China Meteorological Administration, School of Atmospheric Physics, Nanjing University of Information Science & Technology, Nanjing 210044, China

[3]National Center for Atmospheric Research, Boulder, CO 80301, USA

[4]National Meteorological Information Center, China Meteorological Administration (CMA), Beijing 100081, China

[5]Center for Atmospheric and Oceanic Studies, Graduate School of Science, Tohoku University, Sendai, Miyagi 980-8578, Japan

[6]Chinese Academy of Meteorological Sciences, Beijing 100081, China

*Correspondence to*: Chao Liu (chao_liu@nuist.edu.cn)

**Abstract.** Extensive observational and numerical investigations have been performed to better characterize cloud properties. However, due to the large variations of cloud spatiotemporal distributions and physical properties, quantitative depictions of clouds in different atmospheric reanalysis datasets are still highly uncertain, and cloud parameters in the models to produce those datasets remain largely unconstrained. A radiance-based evaluation approach is introduced and performed to assess the quality of cloud properties by directly comparing reanalysis-driven forward radiative transfer results with radiances from satellite observation. The newly developed China Meteorological Administration Reanalysis data (CRA), the ECMWF's Fifth-generation Reanalysis (ERA5), and the Modern-Era Retrospective Analysis for Applications, Version 2 (MERRA-2) are considered in the present study. To avoid the unrealistic assumptions and uncertainties on satellite retrieval algorithms and products, the radiative transfer model (RTM) is used as a bridge to "translate" the reanalysis to corresponding satellite observations. The simulated reflectance and brightness temperatures (BTs) are directly compared with observations from the Advanced Himawari Imager (AHI) onboard the Himawari-8 satellite in the region from 80° E to 160° W between 60° N and 60° S, especially for results over East Asia. Comparisons of the reflectance in the solar and BTs in the infrared (IR) window channels reveal that CRA reanalysis better represents the total cloud cover than the other two reanalysis datasets. The simulated BTs for CRA and ERA5 are close to each other in many pixels, whereas the vertical distributions of cloud properties are significantly different, and ERA5 depicts a better deep convection structure than CRA reanalysis. Comparisons of the BT differences (BTDs) between the simulations and observations suggest that the water clouds are generally overestimated in ERA5 and MERRA-2, whereas the ice cloud is responsible for the overestimation over the center of cyclones in ERA5. Overall, the cloud from CRA, ERA5, and MERRA-



2 show their own advantages in different aspects. The ERA5 reanalysis is found the most capability in representing the cloudy atmosphere over East Asia, and the results in CRA are close to those in ERA5.

**1 Introduction**

As an important element in the Earth atmosphere, clouds play a vital role in the global radiation budget, water cycle, and climate change. Cloud formation is governed by the balance between dynamical, thermodynamic, and microphysical processes (Boucher et al., 2013). Although the representation of cloud in different atmospheric datasets and cloud evolution in regional and global numerical models have been significantly improved in the past few decades (Cess et al., 1989; Cotton

et al, 2003; Arakawa, 2004), cloud is still one of the dominant uncertainties in the atmosphere, and causes difficulties in understanding the energy balance and climate change mechanisms (Dufresne and Bony, 2008; Boucher et al., 2013).

The atmospheric reanalysis, a dataset that combines observations and forecasting products (Dee et al., 2011), provides multivariate records of the global atmospheric circulation, and is widely used in the studies of climate change, cloud property retrieval, and the initialization of numerical modeling. With the advances in computation capability and the improvement of

global observing systems, an increasing number of observed datasets are assimilated into the reanalysis by more advanced data assimilation methods and systems, and the reanalysis is being closer to realistic atmospheres. From the late period of the last century, a series of reanalysis data have been produced, for example, the National Centers for Environmental Prediction (NCEP) 40-yr Reanalysis Project (Kalnay et al., 1996), the 40-year ECMWF Reanalysis (ERA-40; Uppala et al., 2005), the Japanese 25-year Reanalysis (JRA-25; Onogi et al., 2007), the Modern-Era Retrospective Analysis for Research and

Applications (MERRA; Rienecker et al., 2011), the ECMWF's Interim Reanalysis (ERA-Interim; Dee et al., 2011), and the Japanese 55-year Reanalysis (JRA-55; Kobayashi et al., 2015). Though some schemes and systems that support the assimilation of cloud-affected satellite radiance are developed (Chevallier et al., 2004; MaNally, 2009), clouds are difficult to be assimilated into the reanalysis, instead, they are forecasted by numerical weather prediction models (Free et at., 2016). Thus, although many atmospheric parameters in the reanalysis data are increasingly confident, the cloud is still challenging,

and it is important yet difficult to accurately and reasonably assess the cloud properties in the reanalysis.

Because of large advantages of spatial distributions, observed atmosphere from satellite platforms is the best choice in the evaluation of output fields from numerical models. Some previous studies have conducted evaluations of reanalysis or model outputs based on satellite retrieved products. This is known as the satellite- or retrieval-based approach. Interesting results are achieved by this method (Jakob, 1999; Waliser et al., 2009; Hashino et al., 2013), especially for the long-term cloud

cover in the reanalysis. However, satellite retrieval is an inverse solving process. Many assumptions or parameters are needed to infer unknown quantities, and this will introduce some inevitable uncertainties. For example, although the vertical cloud profile is one of the most essential properties in most models, the single homogeneous layer cloud assumption is widely used in most satellite retrieval algorithms for cloud optical and microphysical properties (Wind et al., 2013; Yang et al., 2015), and the artificial assumption will bring many uncertainties in cloud products. Moreover, the scattering properties


of cloud particle model themselves are with lots uncertainties, and they are inconsistence in different retrieval approaches. As a result, due to differences of retrieval algorithms and platforms, the consistence of retrieved products derived from different satellites is still of large challenges (Matsui, et al., 2014), and some evaluations by the satellite-based approach are often questionable.

Therefore, a radiance-based comparison may be a more reasonable choice for cloud properties assessment. In this approach,
simulated radiative parameters, such as brightness temperature (BT) in the infrared (IR) channels or microwave channels and reflectance in the solar channels, can be calculated by a forward radiative transfer model (RTM), and the radiative variables can be directly compared with satellite radiative observations. The RTM helps us build a bridge between model atmospheric parameters (e.g., the reanalysis dataset) and direct satellite observations (Zhang et al., 2019). This will effectively avoid frustration from the uncertainties of satellite retrieval algorithms and products. This approach was first introduced to evaluate
simulated cloud fields in the thermal IR channels by Morcrette (1991) and Yu et al. (1991). With the advantages of confident radiative information and the diversity of satellite channels, the radiance-based method has been applied to evaluate different cloud microphysics schemes (Han et al., 2013), precipitation microphysics schemes (Hashino et al., 2013), and even aerosol properties (Chaboureau et al., 2007), and became an important way to better understand the microphysical and radiative properties of clouds, precipitation, and other atmospheric parameters.

In this study, we extend the application of radiance-based approach to assess the cloud properties from three reanalysis datasets: the China Meteorological Administration Reanalysis (CRA), the ECMWF's Fifth-generation Reanalysis (ERA5; Hersbach and Dee, 2016) and the Modern-Era Retrospective Analysis for Research and Applications, Version 2 (MERRA-2; Gelaro et al., 2017). This is a new aspect to evaluate cloud and atmosphere properties from different atmospheric datasets. The radiative parameters (i.e., BTs in the IR channels and reflectance in the solar channels) from the Advanced Himawari
Imager (AHI) onboard the Himawari-8 satellite (Bessho et al., 2016) are used as the observations.

This paper is organized as follows. The datasets are introduced in Section 2, and the coupled method between cloud microphysical parameters in the reanalysis and optical parameters that are supported by satellite retrieval and RTMs are described in Section 3. In Section 4, we present a brief analysis of the uncertainty of the retrieval-based evaluation. A detailed radiance-based evaluation of cloud properties from the reanalysis, including a case assessment and a long-term
comparison performed with a 36-day dataset (total of 144 realizations) spanning one year, is presented in Section 5. Section 6 summarizes the study.

## 2 Dataset

The newly developed Chinese first-generation atmospheric reanalysis, CRA, is based on the use of the National Oceanic and Atmospheric Administration (NOAA) Global Forecast System (GFS) model and Gridpoint Statistical Interpolation (GSI)
3DVAR data assimilation system (Wu et al., 2002; Kleist et al., 2009) with a T574 spectral resolution (34km grid spacing). The final CRA products will span the period from 1979 to present and is targeted to be produced and released in late 2020.





An interim version of CRA (CRA-interim) for a 10-year period (1 January 2007 – 31 December 2016) at a 6-hourly time interval was produced in February 2018. An abundance data from in-situ observations and multiple satellite instruments, especially for the East Asian regions, have been assimilated into CRA-interim. CRA-interim data used in the study are in 47

pressure levels from the surface to 0.27 hPa with a horizontal resolution of 0.3125° × 0.3125°.

The ERA5 is the latest released numerical dataset of the recent climate. It is currently available for the period from 1979 to present at a 3-hourly time interval, and will be extended from 1950 to present. The Integrated Forecasting System (IFS) Cycle 41r2, is used to assimilate the available observations from satellites and in-situ stations. The spatial resolution of the ERA5 dataset is 0.25° × 0.25°, and the atmospheric data are with 37 pressure levels from the surface to 1 hPa (Hersbach and

Dee, 2016).

The MERRA-2 is the atmospheric reanalysis produced by the Global Modeling and Assimilation Office (GMAO) of the National Aeronautics and Space Administration (NASA), with the Goddard Earth Observing System (GEOS) atmospheric data assimilation system. It provides data from 1980 to present and is designed to build a bridge between the first MERRA reanalysis data and the project's long-term goal of developing an integrated Earth system analysis (IESA; Gelaro et al.,

2017). In this study, the data used from MERRA-2 is at a spatial resolution of 0.5° × 0.625° with 42 levels from the surface to 0.1 hPa.

To compare the quality of the three reanalysis datasets, satellite observed data from Himawari-8 are used. Launched on 7 October, 2014 and operated by the Japan Meteorological Agency (JMA) since 2015, the Himawari-8 is one of the new generation satellite members of the Multi-functional Transport Satellites (MTSATs; Da, 2015; Bessho et al., 2016). The

Advanced Himawari Imager (AHI), which is a radiometer with 16 bands from the solar to IR range, is on board the Himawari-8 to observe the Earth from 80° E to 160° W between 60° N and 60° S. The spatial resolution of the observations is 0.5–2 km and the temporal resolution is 2.5–10 minutes (Iwabuchi et al., 2018). With high spatial and temporal sampling, the AHI measurement is valuable for disaster monitoring and cloud studies, especially for the region over East Asia. Moreover, to provide a clear understanding of the uncertainties and problems from the reanalysis assessment based on the

satellite retrieval approach, two retrieved cloud datasets are chosen for the retrieval-based evaluation. One cloud product is from the solar measurement retrieval. It is based on the AHI reflectance in the 0.64- and 1.6-μm channels, and the Voronoi (Letu et al., 2016) cloud scattering model is utilized in the retrieval (Letu et al., 2018). Another cloud dataset of the Himawari-8 satellite (Iwabuchi et al., 2018) is from the thermal IR measurement. Four channels at 8.6-, 10.4-, 11.2-, and 12.4-μm, which are sensitive to cloud properties such as cloud top height, cloud optical depth, and cloud effective radius are

chosen, and the scattering properties for water and ice particles are from Lorenz-Mie theory (Mie, 1908), and the database of Yang et al. (2013), respectively.

For consistency in the comparison, all datasets used in this study are at a 6-hourly time interval, and the horizontal resolutions are re-gridded by the inverse distance weighted method to match the spatial distribution of the CRA (Guan and Wang, 2007; Holz et al., 2008). An 8-day case and a general comparison with a 36-day dataset (total of 144 realizations)



spanning one year are chosen. Although the size of the evaluated datasets is small, the statistical results are credible, and the significant features are presented.

## 3 Methodology

With our focus on cloudy atmospheres, the accuracy of cloud properties is one of the most critical factors for the reliability of the evaluation. Cloud effective radius and optical depth are key microphysical and optical parameters in determining the

radiation property in each atmospheric layer. However, variables from the reanalysis, e.g., cloud mixing ratio, cannot be directly supported by the fast RTM, and therefore cannot be directly compared with the satellite retrieved cloud optical properties. Thus, a reasonable coupled method between the microphysical properties in the reanalysis and the optical parameters that are comparable to satellite retrieved cloud properties and supported by the RTM is important and challenging. Table 1 lists the geophysical parameters in the reanalysis that are used in our study. A cloud coupled approach with less

empirical or semi-empirical assumptions is performed. In each grid box, the occurrence of cloud or hydrometeor particles is diagnosed with cloud mixing ratio ($q_c$) larger than 0.001 g/kg and the relationship between relative humidity and cloud amount (Slingo, 1980). Ignore the uncertainties caused by the mixed-phase cloud, a temperature threshold of 253 K is used to distinguish cloud phase. If the temperature of cloud layer is larger than 253 K, then the grid box is regarded as a water cloud, otherwise the grid box is regarded as an ice cloud (Mazin, 2004).

The effective radius ($R_w$) in each water cloud grid box is approximated by the cloud mixing ratio ($q_c$) and number concentration ($N_w$) (Thompson et al., 2004):

$$R_w = \frac{1}{2} \times (\frac{6\rho_a q_c}{\pi \rho_w N_w})^{\frac{1}{3}} \tag{1}$$

where $\rho_a$ is the density of air, which is determined by the pressure and temperature in the corresponding layer. The density of water cloud particles ($\rho_w$) is assumed to be 1000 kg/m$^3$. The water cloud number concentration of $N_w = 3 \times 10^8$ m$^{-3}$ is

assumed over the continent and $N_w = 1 \times 10^8$ m$^{-3}$ is used over the ocean region (Miles et al., 2000).

The ice cloud effective radius ($R_i$) is obtained by the relationship between mass extinction coefficient ($k$) and cloud effective radius. The $k$ can be given by an empirical relationship based on in-situ measurements (Heymsfield and McFarquhar, 1996; Platt, 1997; Heymsfield et al., 2003):

$$k = 0.018 \times (\text{IWC})^{-0.14} \tag{2}$$

where IWC is the corresponding ice water content, and it is obtained from the cloud mixing ratio and density of air. Once $k$ is produced, the corresponding $R_i$ can be available from the cloud property database.

The optical depth determines the attenuation of radiation in the cloud layer. When the cloud effective radius ($R_w$ or $R_i$) and the corresponding $k$ are given, the cloud optical depth ($\tau$) in the visible wavelength can be obtained by:

$$\tau = k \times \text{CWP} \tag{3}$$

where CWP is the cloud (water or ice phase cloud) water path in each grid box and is found by integrating the cloud water content (CWC) from the cloud base ($h_{base}$) to top ($h_{top}$):





$$\text{CWP} = \int_{h_{base}}^{h_{top}} \text{CWC} \, dh \tag{4}$$

Then the cloud optical depth can be directly compared with the satellite retrieved cloud optical depth, and be converted into the corresponding optical depth at a specific wavelength when performing the RTM simulations.

## 4 Retrieval-based evaluation

As we know, the equivalent single cloud structure is widely used in the satellite retrieval algorithms, and retrieved cloud products from different algorithms usually have some inconsistences and uncertainties. The assessment of clouds based on retrieval-based evaluation approach is controversial. Thus, we first perform an evaluation of cloud properties from reanalysis based on two Himawari-8 satellite retrieved cloud products.

Figure 1 shows the spatial distribution of the cloud optical depth, effective radius, and cloud top temperature (CTT) from two satellite retrieved cloud datasets (i.e., from solar and thermal infrared measurements), the CRA, ERA5, and MERRA-2, and the results are from 00:00 UTC on 12 September 2016. To obtain the equivalent cloud properties of the reanalysis, the cloud optical depth ($\tau(l)$) from the surface ($l=0$) to the top of atmosphere (TOA) ($l=s$) is summed as the column cloud optical depth ($\tau_{column}$) in every grid:

$$\tau_{column} = \sum_{l=1}^{l=s} \tau(l) \tag{5}$$

The column cloud effective radius ($R_{column}$) is calculated by integrating the effective radius in each layer ($R(l)$), with the corresponding optical depth ($\tau(l)$) as the weighting coefficient:

$$R_{column} = \frac{\sum_{l=0}^{l=s}(\tau(l) \times R(l))}{\sum_{l=0}^{l=s} \tau(l)} \tag{6}$$

For the definition of CTT, if the integrated cloud optical depth from the TOA to layer $l$ satisfies the threshold of 0.1, then the corresponding temperature of layer $l$ is considered as the CTT.

The differences between the satellite retrieved cloud properties from the solar and thermal IR measurements indicate the significant uncertainties in satellite retrieved products. Because it is difficult to provide quantitative cloud optical depth retrieval for high and thick clouds with large cloud optical depth for the thermal IR measurement, the values for many pixels are smaller than those retrieved from the solar measurement (Yang et al., 2015). Compared with the satellite retrieved products, the cloud optical depth simulated by atmospheric reanalysis is extremely large, especially for those from MERRA-2, and the effective radius in corresponding pixel is extremely small. Although the CTT distributions between CRA reanalysis and satellite retrieval from the solar measurement are better than other comparisons, the correlation between the two is small.

Figure 2 gives the quantitative pixel-to-pixel comparison of the results from 10 September 2016 to 17 September 2016, and more than one million cloud pixels are considered. The color contours indicate the occurrence of cloud properties from satellite retrieval and the CRA reanalysis. We notice that the logarithmic values of the cloud optical depth are computed for the comparison of cloud optical depth, thus, some values are in the negative range. Compared with the retrieved results from the solar or IR instruments, the cloud effective radius in CRA are extremely smaller over most pixels with thick clouds, but





are larger over pixels with thin cloud optical depth, therefore, a significant hierarchical structure is shown for the pixel-to-pixel comparison of cloud effective radius. The correlation coefficients for cloud optical depth, effective radius, and CTT between the CRA reanalysis and satellite retrieval are all smaller than 0.5.

Figures 1 and 2 illustrates that because of the differences among different retrieval algorithms, the satellite retrieved cloud products themselves are with lots of uncertainties. Moreover, it is hard to give a significant and quantitative assessment of the reanalysis based on satellite retrieved products, regardless of whether the products are from solar or thermal infrared measurements. The retrieval-based evaluation itself is in arguable, and it is not a reasonable and reliable approach to assess the cloud properties form reanalysis and other similar modeled cloud products.

## 5 Radiance-based evaluation

In the radiance-based evaluation, the Community RTM (CRTM) is used to calculate satellite observed radiative variables based on the synthetic atmospheric variables in the reanalysis. The CRTM is designed to simulate radiance and radiance gradients at the (TOA), and has been widely applied in radiance assimilation, remote sensing sensor calibration, climate reanalysis and so on. Procedures for solving the radiative transfer in the model are divided into various independent modules (e.g., gaseous absorption module, surface emissivity module, and cloud absorption/scattering module) (Chen et al., 2008; Ding et al., 2010). To improve the computational efficiency, the advanced fast adding-doubling method (ADA) method is used (Liu and Weng, 2006), and it is 1.7 times faster than the vector discrete ordinate method (Weng, 1992) and 61 times faster than the classical adding-doubling method (Twomey et al., 1966; Hansen and Hovenier, 1971). Four major surface types (i.e., water, land, ice, and snow) are included in the surface emissivity module, and the corresponding spectral library from visible to microwave wavelengths is pre-prepared for the emissivity calculation (Chen et al., 2008; Baldridge et al., 2009).

To minimize the numerical errors and uncertainties from radiative transfer computation, the cloud optical property look-up tables (LUTs) in the absorption/scattering module of CRTM are optimized before the simulation. We recalculate the single-scattering optical properties of water clouds by Lorenz-Mie theory (Mie, 1908), and the single-scattering optical properties of ice clouds are based on the data library developed by Yang et al. (2013). A gamma size distribution with an effective variance of 0.1 (Hansen and Travis, 1974) is assumed to compute the bulk scattering properties (i.e., the extinction coefficient, single-scattering albedo, asymmetry factor and phase function coefficients). Comparisons between the simulation and observation show that the CRTM with new cloud optical property LUTs substantially improves the simulation on cloudy atmospheres (Yi et al., 2016; Yao et al., 2018).

To obtain the most realistic representation of the radiance from the TOA, the full layer atmospheric profiles (i.e., the pressure, temperature, and water vapor) and cloud optical properties that are computed in Section 3 are directly kept and adopted by the CRTM for the calculation of gas absorption and emission, and cloud scattering. The surface characteristics (e.g., surface type, altitude, and surface temperature) are also necessary for the CRTM to give the surface radiative property.





Because the ozone absorption is insensitive in the channels of interest, the climatological ozone profiles are used in the simulation.

**5.1 Case assessment**

We first present a comparing study spanning an 8-day typical case to provide a detailed assessment of the cloud properties
from three reanalysis datasets From 10 to 17 September 2016, the super typhoon Meranti, which is one of the most powerful tropical cyclones on record, was monitored. The extremely favorable atmospheric environment, including adequate water vapor, increased outflow in the upper layer, and unusually warm sea surface temperature, intensified the structure and energy of the typhoon. Meanwhile, on 11 September 2016, another tropical depression was detected and monitored over the Northwest Pacific Ocean, and it evolved into the typhoon Malakas on 13 September. The interaction of the two typhoons
increased the water vapor transportation, which promoted the development of deeper and thicker clouds, and the rapid enhancement of the typhoons (Zhou and Gao, 2016). When Meranti passed over the Philippines and China, it produced heavy rain and hurricane-force winds and caused extensive damage.

Figure 3 shows the spatial distribution patterns of the reflectance in the 0.64- and 1.6-μm channels. The observed and simulated results are taken at 00:00 UTC on 12 September 2016. Four typical regions marked by red boxes are chosen for
better understanding and illustration. Because the channel in the visible wavelength (0.64-μm) is non-absorbing, the reflectance is primarily constrained by the cloud optical depth. Therefore, some cloud macro characteristics can be recognized from the result in this channel. The pixels with reflectance close to 1 (the whiter points) indicate the region covered by optically thick clouds. A qualitative comparison between the observation and the simulation shows that the results for CRA reanalysis more reasonably represent the cloud spatial distribution than those for the other two reanalysis
datasets. The simulations from ERA5 and MERRA-2 obviously overestimate the cloud cover, and the overestimated cloud pixels are mostly over the ocean regions, for example, the region B, C and surrounding areas. In the 1.6-μm channel, the radiance observed from the TOA is significantly different for different phase clouds. Because ice particles have a stronger absorption property, the reflectance in this channel is usually smaller for pixels covered by ice clouds than those covered by water clouds. Thus, we can give a general distinction of the cloud phase based on the information in this shortwave IR
channel. The similar characteristics between the observation and simulation over region A indicate that CRA, ERA5, and MERRA-2 all have capabilities to distinguish ice and water phase clouds. Comparing the results over region B, the three reanalysis datasets all represent the cloud phase characteristics of the cyclones. More pixels with larger reflectance values for ERA5 suggest that although the cloud distributions in ERA5 and MERRA-2 are both overestimated over region B, the causes for the results are different. Some overestimated clouds in ERA5 reanalysis are from water phase clouds, whereas
they are mostly caused by ice phase clouds in MERRA-2. For pixels over regions C and D, the overestimation comes from the water cloud in ERA5 and MERRA-2.

Different from the reflectance in the solar channels, the BTs in the IR channels are available for both daytime and nighttime. For further assessment and comparison, the discussion below is mostly based on the results in the three IR channels (one is





in the water vapor channel, and two are in the window channels), and the time period is the same as that in Figure 3. Figure 4

illustrates the observed BTs in the 6.2-, 8.6-, and 11.2-μm channels, and the brightness temperature differences (BTDs) between the simulated BTs from CRA, ERA5, and MERRA-2 and the observations. The corresponding statistical error parameters (i.e., the mean, maximum, minimum, and standard deviation of BTDs) are listed in Table 2. Some typical regions and pixels are marked by boxes and dashed lines for better understanding and analysis. The IR window channels (8- to 12-μm) have less molecular absorption, and they are mostly sensitive to the surface temperature and cloud profiles. Therefore,

the BTs in these channels are usually used to evaluate cloud properties or surface temperature (King et al., 1992; Mao et al., 2005). In the 6.2-μm channel, because of large sensitivity to a broad upper-layer humidity, the BTs are used to infer the mid- to high-layer water vapor content. Similar horizontal distributions between the observation and simulation in the two window channels generally confirm the dependable capabilities of the three reanalysis datasets to represent the atmospheric characteristics on cloudy and clear-sky. Over the entire region, the smallest average error of -1.59 K in the 11.2-μm channel

indicates the best simulated BTs for ERA5, and the average results for CRA are close to it. However, the simulated error is much larger for MERRA-2, and the mean BTD is -9.19 K. Region A (i.e., the continental region) is characterized by low-layer clouds or clear-sky conditions, with a mean BTs of 268.55, 270.12, 269.57, and 263.21 K for the observation, CRA, ERA5, and MERRA-2. The slightly underestimated cloud optical depth or cloud top height over this region may cause positive mean BTDs of 1.56 and 1.01 K for CRA and ERA5, respectively. However, the negative mean BTDs indicate that

the properties are overestimated in the simulation for MERRA-2. Meanwhile, we need to note that some other atmospheric or surface properties may also cause similar results because of the uncertain and complex terrain features over the arid or semiarid regions. For the Tibetan Plateau, the limitation of the in-site observations results in uncertainties for the reanalysis datasets. Compared with the continental regions, larger simulation errors over ocean are primarily associated with more complex cloud distributions and structures. Over region B, broad simulated clouds with BTs between 220 and 250 K are

largely responsible for the negative mean BTDs. The absolute BTD may reach as large as 80–90 K in the window channel, and it is almost 15–20 K larger than that over region A. More series excessive cloud pixels for MERRA-2 reanalysis explain the mean BTD of -19.02 K in the 11.2-μm channel. The negative mean BTDs over region B for CRA and MERRA-2 in the 6.2-μm channel suggest the excessive integrated mid- to high-layer water vapor content. The positive mean BTD for ERA5 over region B in this water vapor channel reveals a general insufficient water vapor content over the corresponding layer,

and this results in the underestimation of upper-layer clouds. Meanwhile, the mean BTD of -2.35 K in the 11.2-μm channel indicates that the overestimation of clouds should be related to low- or mid-layer clouds in this region. However, more water vapor content is represented in ERA5 over region C than in CRA and MERRA-2, and it is closer to the realistic atmosphere. Compared with the observation, a similar cyclone structure is captured in the imagery of IR window channel.

To give a quantitative evaluation of the results in Figure 3, the pixel-to-pixel comparisons over the entire region are shown in

Figure 5. The color contours show the occurrence of the reflectance from the observations and simulations, and the color bar is shown on a logarithmic scale. The symmetry distribution with the high occurrence frequency following around the black 1:1 line for the results of CRA reveals a better agreement with the observed reflectance than the ERA5 and MERRA-2. The



correlation coefficients of 0.66 and 0.62 for CRA in the 0.64- and 1.6-μm channels, respectively, reveal the best simulation in the solar channels. The simulations for ERA5 and MERRA-2 are clearly higher than the observations in some pixels,

which yield a secondary high occurrence frequency band over the observed reflectance less than 0.2. This band corresponds to the overestimated cloud distributions in Figure 3. The correlation coefficients for ERA5 and MERRA-2 are 0.65 and 0.53, respectively, in the 0.64-μm channel, and they are less than 0.5 in the 1.6-μm channel.

Figure 6 gives a similar pixel-to-pixel evaluation, but it is for the results in the IR channels. The correlation coefficients are all larger than 0.6, and the high occurrence is around the 1:1 line, revealing good agreements between the simulated and

observed BTs in the 11.2-, 8.6-, and 6.2-μm channels, especially for CRA and ERA5.

To further demonstrate a quantitative evaluation of the results in the solar and IR channels, Figure 7 shows the probability (top panels) and cumulative probability (bottom panels) for the simulations and observations. The total cloud is obviously overestimated in the results from ERA5 and MERRA-2, and the probability density of reflectance larger than 0.2 in the 0.64-μm channel is larger than that in the Himawari-8 observation. In the IR window channels, the simulation from MERRA-2

overestimates the probability density against the observation between 220 and 275 K, reflecting the overestimation of low- and mid-layer cloud. A boundary in the BT of 250 K indicates the overestimation of the mid-layer clouds and slight underestimation of the low-layer clouds for CRA reanalysis. For ERA5, the low-layer clouds are overestimated, but the mid- and high-layer clouds are underestimated, especially for clouds with a top temperature less than 230 K. Similar probability density structures between simulations for ERA5 and the observation in the 6.2-μm channel reveal a more reasonable water

vapor distribution over the entire region, compared to those of the other two reanalysis datasets. Matusi et al. (2014) point out that the cumulative probability density is a better metric to assess the cloud cover than satellite cloud products. When a threshold of BT approximately 280 K in the 11.2-μm channel is assumed to be present of cloud pixels, the simulated cloud cover for CRA reanalysis achieves the best agreement with the observation. However, the cloud cover is overestimated by 17% and 33% in ERA5 and MERRA-2, respectively.

The atmospheric and cloud profiles (i.e., temperature, cloud mixing ratio, cloud effective radius and optical depth) over pixels of 18°N (marked by blue dashed lines in Figure 4) are shown in Figure 8, and the corresponding integrated cloud mixing ratio, cloud optical depth and the number of cloud layers in each column are illustrated in Figure 9. Compared with the differences in the temperature profiles, the differences in the cloud mixing ratio profiles are more conspicuous among the three reanalysis. The cloud mixing ratio is insufficient over the low-to-mid layer in CRA, but in ERA5 reanalysis, the

shortage is over the mid-to-high layer. This directly results in the differences in cloud vertical structures. On the one hand, in Figure 9, the integrated cloud properties cover up the inconsistency, and they are close to each other and result in similar simulated BTs in thin cloud pixels. On the other hand, similar integrated properties may cause significantly different BTs. Although the number of cloud layers and the integrated cloud optical depth are close in some pixels over region D (Figure 4), the simulated BTs in the 11.2-μm channel are much lower for CRA than for ERA5 reanalysis. This is caused by an abnormal

excessive cloud mixing ratio or optical depth in the mid to high-layer. For MERRA-2 reanalysis, the widespread cloud mixing ratio brings in overestimated integrated cloud optical depth and cloud distributions in many pixels.





Different spectral channels have their own sensitivities to atmospheric and cloudy properties, so different cloud properties or atmospheric conditions can be detected and validated by the BTDs among different channels (Baum et al. 2000; Otkin, et al. 2009). Different from previous analysis based on single channel results, Figure 10 shows the observed and simulated BTDs of 11.2–12.4-μm, 8.6–11.2-μm, and 6.2–11.2-μm. The absorption of atmospheric water vapor in the 12.4-μm channel is greater than that in the 11.2-μm channel, and BTDs for 11.2–12.4-μm are usually positive in most regions. The cloud emissivity increases as the optical depth increases, which weakens the influence from the atmosphere below the cirrus clouds, and results in similar BTs in the two channels. Thus, smaller or zero BTDs are detected across the deep convective region (e.g., region C) and thick cloud regions. Meanwhile, because of the enhanced extinction of small ice particles in the 12.4-μm channel, the BTDs for thinner clouds around thick cloud pixels are large. Although the absorptive properties for different phase particles are similar in the 8.6-μm channel, the absorption for ice clouds is larger than that for water clouds in the 11.2-μm channel. Thus, the BTDs of 8.6–11.2-μm are positive for ice clouds and negative for water clouds in a typical case. Comparing the BTDs in the particular cloud region (e.g., region B), simulations for CRA are close to the observations, and the mean BTDs for them are 0.16 and 0.14 K, respectively. The negative mean BTDs in this region for ERA5 and MERRA-2 indicate the overestimation of water clouds or some underestimation of ice clouds. Because of the strong water vapor in the 6.2-μm channel and the negative temperature lapse rate in the troposphere, the BTDs of 6.2–11.2-μm are usually negative, and increase as the cloud height increases. The largest negative BTDs are often in the clear-sky region with sufficient water vapor and high surface temperature, and the positive or near zero BTDs correspond to overshooting cloud tops. Although the simulation for ERA5 reanalysis generally underestimates the mid to high-layer water vapor content and upper-layer cloud in the entire and B region, as we mentioned before, if we isolate the overshooting cloud top by BTDs less than 0 K, the ERA5 has the closest structure and distribution to the observation over the three reanalysis datasets, corresponding to the analysis of region C.

Ratios of the simulation-to-observation frequency of pixels with particular BTs in the 11.2-μm channel are illustrated in Figure 11 to give a comprehensive evaluation for the cloud cover in the 8-day case. A threshold of BT between 255 and 280 K is used to infer the low-layer clouds, and the pixels with mid-layer clouds are represented by BT between 220 and 255 K. The high-layer clouds are classified by BT less than 220 K, and the demarcation between cloudy and clear-sky is 280 K. The changes in the cloud amount in different layers are small during this particular case. The mean ratio of 1.00 for CRA demonstrates an excellent simulated TCC in the three reanalysis datasets. Although the simulated high-layer cloud ratio for MERRA-2 reanalysis is reasonable, the excessive mid- and low-layer clouds together result in a widespread overestimated TCC. For ERA5 reanalysis, the ratio of 1.09 in the mid-layer clouds reveals a better simulation than that for CRA and MERRA-2. Ratio of approximately 0.5 suggests a large underestimation of the high-layer clouds, whereas the shortage is covered up by the overestimated low-layer clouds, and it contributes to the overestimation of the total cloud cover.



**5.2 Long-term assessment**

Further, a dataset spanning in 2016, with a total of 144 realizations (the realizations are from the 5[th], 15[th], and 25[th] of each

month, and 4 data in a 6-hourly time interval in each day are available) for each reanalysis is chosen to give a generally long-term comparison and assessment. Although the size of the dataset is not large enough, the significant characteristics are presented.

Figure 12 gives the ratio of clouds in different layers, and the definition and classification are the same as those in Figure 11, and the average values are listed in Table 3. For CRA and ERA5, the ratios of clouds in different layers show relatively weak

variation over time, and the variation ranges and mean values are similar to the results in Figure 11. However, the results for MERRA-2 are with obvious seasonal variation characteristics. The simulated mid-, high-layer and total cloud ratios in summer are significantly larger than those in other seasons. This is associated with the widespread overestimated cloud distributions in MERRA-2 reanalysis, and the more frequent convective systems with thicker and higher clouds in summer aggravate the excessive overestimation.

Figure 13 illustrates the average BTDs between the simulations from CRA, ERA5, and MERRA-2 and the observations in the IR window and water vapor channels. Over the entire region, most pixels with average BTDs around 0 K in the IR window channels reveal a general good simulation from CRA and ERA5. Regions with larger deviations are generally over the arid or semiarid areas (as marked in region A in Figure A), and the surrounding regions of the equator. For MERRA-2, the significant deviations with negative BTDs are over the Intertropical Convergence Zone (ITCZ), and the phenomenon is

extended to the region around 20° N. Most pixels of positive BTDs in the water vapor channel for ERA5 indicate an underestimation of water vapor, and it is more obvious over the region of ITCZ.

Figure 14 shows the temporal variation of the mean error (MBTD), standard error (SBTD), and correlation coefficient (R) in the 11.2-, 8.6-, and 6.2-μm channels, and the corresponding average values are listed in Table 3, together with the results for Figure 12. Three statistical parameters show seasonal variation characteristics over time and the largest errors are in summer

because of more complex weather systems and clouds. The mean errors for the three reanalysis datasets are always negative in the IR window channels, demonstrating the general overestimation of clouds, especially for the results in MERRA-2 reanalysis. In the 6.2-μm channel, the opposite phases of mean errors indicate the general underestimated mid to high-layer water vapor for ERA5 but an overestimation for CRA, corresponding to the analysis in Figure 13.

Overall, the spatial distributions of the average BTDs in Figure 13 and the statistical evaluation in Figure 14 indicate that the

results for ERA5 have the best generalizable capability to represent atmospheric and cloud characteristics over the corresponding large region of the Himawari-8 observation, with the smallest absolute mean error of 0.92 K, the smallest standard error of 12.77 K, and the largest correlation coefficient of 0.80. The results in CRA are close to those in ERA, whereas in MERRA-2, the deviations are slightly larger. Large and systemic deviations for the three reanalysis are mostly over the oceanic region around the equator and areas with complex surface features. The atmospheric and cloud



characteristics are complex and volatile, and the in-site observations are limited over these regions. The atmospheric and cloud in the reanalysis are with lots of uncertainties.

**6 Summary**

This study performs an assessment of cloud properties from three reanalysis datasets (i.e., the CRA, ERA5, and MERRA-2) with the Himawari-8 satellite observation by a radiance-based approach. The atmospheric and cloud variables in the

reanalysis are converted into BTs or reflectance, with the help of a reasonable cloud and atmosphere coupled method and the widely used forward RTM (i.e., CRTM), and they are compared and analysed with the satellite direct observations.

The assessment indicates that the atmospheric and cloud characteristics from CRA, ERA5, and MERRA-2 are mostly depicted. The BTs in the IR window channels (i.e., 11.2- and 8.6-μm) and reflectance in the 0.64-μm channel reveal the excellent TCC in CRA. ERA5 reanalysis has the most reasonable mid-layer clouds, but large underestimation of high-layer

clouds. However, the shortage may be covered up by the overestimated low-layer clouds, and this results in an overestimation of TCC. For the results in MERRA-2, the high-layer clouds are more reasonable than clouds over other layers, and the widespread overestimated TCC is mostly caused by the overestimation of low- and mid-layer clouds. From the results in the 6.2-μm channel, obvious overestimated mid to high-layer water vapor is shown in CRA and MERRA-2, whereas it is underestimated in ERA5 over most regions. The BTD comparisons of 6.2–11.2-μm suggest that ERA5 has the

most reasonable overshooting cloud top structures and distributions. The reflectance in the 1.6-μm channel and the BTDs of 8.6–11.2-μm reflect the overestimated water vapor pixels over the ocean region in ERA5 and MERRA-2. However, it is slightly different over the center of the cyclone because more ice cloud pixels are depicted in ERA5 reanalysis.

Generally, the CRA, ERA5, and MERRA-2 are all capable of representing the atmospheric and cloud characteristics over the Himawari-8 observed region. Seasonal variation features over time are shown in a long-time assessment. The larger

statistical errors occur over the oceanic region around the equator and areas with complex surface features, because of the complex atmospheric and cloud structures, and the limitation of in-site observations that can be assimilated into the reanalysis. The largest correlation coefficients of 0.80 and 0.90 between the simulations and observations in the IR window and water vapor channels, respectively, demonstrate that the ERA5 reanalysis achieves the best simulations. The results for CRA also reveal reasonable simulations, and they are close to those in ERA5, whereas for MERRA-2 reanalysis, the

deviations are slightly larger.

Compared with the assessment by satellite retrieved cloud products, the feasible direct comparison of radiative parameters provides a more reasonable evaluation of the microphysical and radiative properties of the atmospheric and cloud properties from the reanalysis. It effectively avoids many uncertainties associated with satellite retrieved products, such as the scattering properties of cloud model, retrieval algorithms, and platforms, and more interesting results and information are

obtained. Although the discussion in this manuscript is focus on the observed region of Himawari-8 satellite on cloudy atmosphere, this approach can be applied to perform the evaluation of more parameters (e.g., cloud, aerosol, precipitation, and so on) from different atmospheric datasets or modeled results. More reasonable analysis and interested information





should be investigated and detected, and it should have a chance to contribute to the improvement of cloud properties in regional or global models and the designation of observations.

**Data availability.**

The data in this study are available at https://github.com/carrolyb/Data_Cloud_Evaluation_2019/.

**Author Contributions.**

BY and CL designed the study, carried out the research, and performed data analysis. BY, CL, YY, ZL, CS, HI, and FW discussed the results and wrote the paper. All authors gave approval for the final version of the paper.

**Competing interests.**

The authors declare that they have no conflict of interest.

**Acknowledgements.**

The CRTM is provided by the Joint Center for Satellite Data Assimilation (JCSDA, ftp://ftp.emc.ncep.noaa.gov/jcsda/CRTM). The CRA dataset is provided by the National Meteorological Information Center
of China Meteorological Administration (CMA). The ERA5 and MERRA-2 reanalysis datasets are available at https://www.ecmwf.int/en/forecasts/datasets/archive-datasets/reanalysis-datasets/era5, and https://gmao.gsfc.nasa.gov/reanalysis/MERRA-2, respectively, The observation of Himawari-8 satellite and retrieved cloud dataset from solar measurement are from https://www.eorc.jaxa.jp. The cloud products retrieved from the thermal infrared measurement is supported by Dr. Hironobu Iwabuchi. We acknowledge funding support by the National Natural Science
Foundation of China (NSFS, grants 41571348 and 41590873), the Special Fund for Meteorological Scientific Research in Public Interest (GYHY 201506002), and the Postgraduate Research & Practice Innovation Program of Jiangsu Province (KYCX18_1004). The computation is supported by the National Supercomputer Center in Guangzhou (NSCC-GZ).

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





**Table 1.** Geophysical parameters from the reanalysis datasets used in the assessment.

| Ordinal | Parameters |
| --- | --- |
| 1 | Temperature at surface |
| 2 | Pressure at surface |
| 3 | Cloud mixing ratio |
| 4 | Atmospheric profiles (pressure, specific humidity, temperature, height) |




**Table 2.** Mean, maximum, minimum, and standard deviation of brightness temperature differences between the simulated and observed BTs (simulation−observation) in Figure 4.

| 11.2-μm | All | | | Region A | | | Region B | | | Region C | | | Region D | | |
|---|---|---|---|---|---|---|---|---|---|---|---|---|---|---|---|
| | CRA | ERA5 | MERR A-2 | CRA | ERA5 | MERR A-2 | CRA | ERA5 | MERR A-2 | CRA | ERA5 | MERR A-2 | CRA | ERA5 | MERR A-2 |
| Mean | -2.36 | -1.59 | -9.19 | 1.56 | 1.00 | -5.35 | -4.21 | -2.35 | -19.02 | 13.93 | 1.40 | -1.59 | -10.30 | -0.97 | -9.60 |
| Max | 94.27 | 79.95 | 79.68 | 79.68 | 62.43 | 67.82 | 94.27 | 79.39 | 74.73 | 87.57 | 53.82 | 48.63 | 85.28 | 66.80 | 67.74 |
| Min | -81.81 | -88.45 | -55.75 | -55.75 | -39.83 | -59.34 | -81.37 | -88.45 | -82.36 | -64.75 | -61.08 | -75.32 | -75.32 | -70.00 | -74.53 |
| Std | 15.92 | 12.16 | 17.75 | 16.21 | 10.93 | 14.07 | 22.66 | 16.83 | 24.28 | 32.06 | 21.46 | 28.47 | 22.86 | 15.17 | 20.39 |

| 8.6-μm | All | | | Region A | | | Region B | | | Region C | | | Region D | | |
|---|---|---|---|---|---|---|---|---|---|---|---|---|---|---|---|
| | CRA | ERA5 | MERR A-2 | CRA | ERA5 | MERR A-2 | CRA | ERA5 | MERR A-2 | CRA | ERA5 | MERR A-2 | CRA | ERA5 | MERR A-2 |
| Mean | -2.40 | -2.31 | -9.32 | 1.80 | 0.58 | -3.31 | -4.18 | -3.40 | 19.34 | 13.22 | 0.21 | -2.80 | -9.77 | -2.39 | -10.40 |
| Max | 94.57 | 77.96 | 74.23 | 77.01 | 59.82 | 63.74 | 94.57 | 77.96 | 72.71 | 87.98 | 53.56 | 45.82 | 83.53 | 64.82 | 63.41 |
| Min | -77.79 | -88.39 | -80.95 | -52.29 | -38.31 | -56.42 | -77.39 | -88.39 | -80.95 | -64.92 | -59.28 | -69.40 | -73.16 | -61.79 | -74.22 |
| Std | 15.30 | 11.62 | 17.05 | 15.24 | 10.44 | 12.92 | 21.98 | 16.26 | 23.50 | 32.79 | 21.89 | 28.76 | 21.57 | 14.42 | 19.17 |

| 6.2-μm | All | | | Region A | | | Region B | | | Region C | | | Region D | | |
|---|---|---|---|---|---|---|---|---|---|---|---|---|---|---|---|
| | CRA | ERA5 | MERR A-2 | CRA | ERA5 | MERR A-2 | CRA | ERA5 | MERR A-2 | CRA | ERA5 | MERR A-2 | CRA | ERA5 | MERR A-2 |
| Mean | -0.81 | 0.71 | -1.19 | -0.98 | 0.00 | -0.25 | -0.94 | 1.80 | -2.77 | 8.29 | 4.14 | 4.92 | -3.55 | 1.07 | -1.43 |
| Max | 41.48 | 43.14 | 36.86 | 31.16 | 29.30 | 29.50 | 41.48 | 43.14 | 36.86 | 37.24 | 31.72 | 33.36 | 29.60 | 29.13 | 35.37 |
| Min | -26.61 | -39.80 | -29.49 | -12.08 | -7.59 | -13.90 | -26.61 | -39.80 | -29.49 | -19.43 | -25.75 | -22.32 | -24.81 | -19.14 | -25.15 |
| Std | 4.24 | 3.51 | 4.89 | 3.46 | 2.70 | 3.44 | 6.76 | 5.44 | 7.57 | 12.78 | 9.47 | 12.50 | 6.86 | 4.72 | 6.37 |





**Table 3.** Average ratios of cloud pixels for different layer clouds in Figure 12, and the average values of the mean error (MBTD), standard error (SBTD), and correlation coefficient (R) between the observations and simulations in Figure 14.

| Cloud | All | | | Spring | | | Summer | | | Autumn | | | Winter | | |
|---|---|---|---|---|---|---|---|---|---|---|---|---|---|---|---|
| | CRA | ERA5 | MERR A-2 | CRA | ERA5 | MERR A-2 | CRA | ERA5 | MERR A-2 | CRA | ERA5 | MERR A-2 | CRA | ERA5 | MERR A-2 |
| Total | 1.02 | 1.11 | 1.27 | 1.03 | 1.11 | 1.27 | 1.01 | 1.14 | 1.35 | 1.00 | 1.11 | 1.26 | 1.03 | 1.09 | 1.20 |
| Low | 0.89 | 1.21 | 1.11 | 0.90 | 1.19 | 1.08 | 0.87 | 1.22 | 1.12 | 0.89 | 1.20 | 1.09 | 0.92 | 1.21 | 1.13 |
| Mid | 1.23 | 1.04 | 1.61 | 1.27 | 1.04 | 1.66 | 1.27 | 1.09 | 1.81 | 1.22 | 1.02 | 1.61 | 1.17 | 0.99 | 1.35 |
| High | 1.03 | 0.39 | 0.87 | 1.15 | 0.38 | 0.94 | 1.03 | 0.39 | 0.99 | 0.87 | 0.38 | 0.79 | 1.08 | 0.41 | 0.75 |
| 11.2-µm | | | | | | | | | | | | | | | |
| MBTD | -2.08 | -0.92 | -7.45 | 2.53 | -0.89 | -7.53 | -2.08 | -1.29 | -9.58 | -1.84 | -0.94 | -7.43 | -2.07 | -0.56 | -5.26 |
| SBTD | 15.66 | 12.77 | 17.53 | 15.53 | 12.69 | 17.29 | 16.39 | 13.42 | 19.24 | 14.83 | 12.84 | 17.35 | 14.92 | 12.14 | 16.24 |
| R | 0.75 | 0.80 | 0.65 | 0.75 | 0.80 | 0.66 | 0.72 | 0.78 | 0.59 | 0.74 | 0.80 | 0.66 | 0.77 | 0.83 | 0.70 |
| 8.6-µm | | | | | | | | | | | | | | | |
| MBTD | -2.20 | -1.68 | -7.83 | -2.44 | -1.61 | -7.86 | -2.26 | -2.17 | -9.97 | -1.93 | -1.68 | -7.80 | -2.16 | -1.27 | -5.68 |
| SBTD | 14.98 | 12.16 | 16.75 | 14.82 | 12.07 | 16.48 | 15.70 | 12.79 | 18.36 | 15.14 | 12.25 | 16.62 | 14.27 | 11.53 | 15.54 |
| R | 0.75 | 0.81 | 0.65 | 0.75 | 0.81 | 0.66 | 0.73 | 0.78 | 0.59 | 0.74 | 0.80 | 0.66 | 0.77 | 0.83 | 0.70 |
| 6.2-µm | | | | | | | | | | | | | | | |
| MBTD | -0.78 | 0.78 | -0.92 | -0.83 | 0.75 | -1.00 | -0.85 | 0.73 | -1.37 | -0.70 | 0.82 | -0.86 | -0.73 | 0.84 | -0.45 |
| SBTD | 4.43 | 3.78 | 5.05 | 4.36 | 3.68 | 4.94 | 4.53 | 3.96 | 5.54 | 4.41 | 3.80 | 4.98 | 4.40 | 3.67 | 4.75 |
| R | 0.87 | 0.90 | 0.82 | 0.87 | 0.90 | 0.82 | 0.87 | 0.89 | 0.79 | 0.87 | 0.90 | 0.83 | 0.88 | 0.91 | 0.86 |



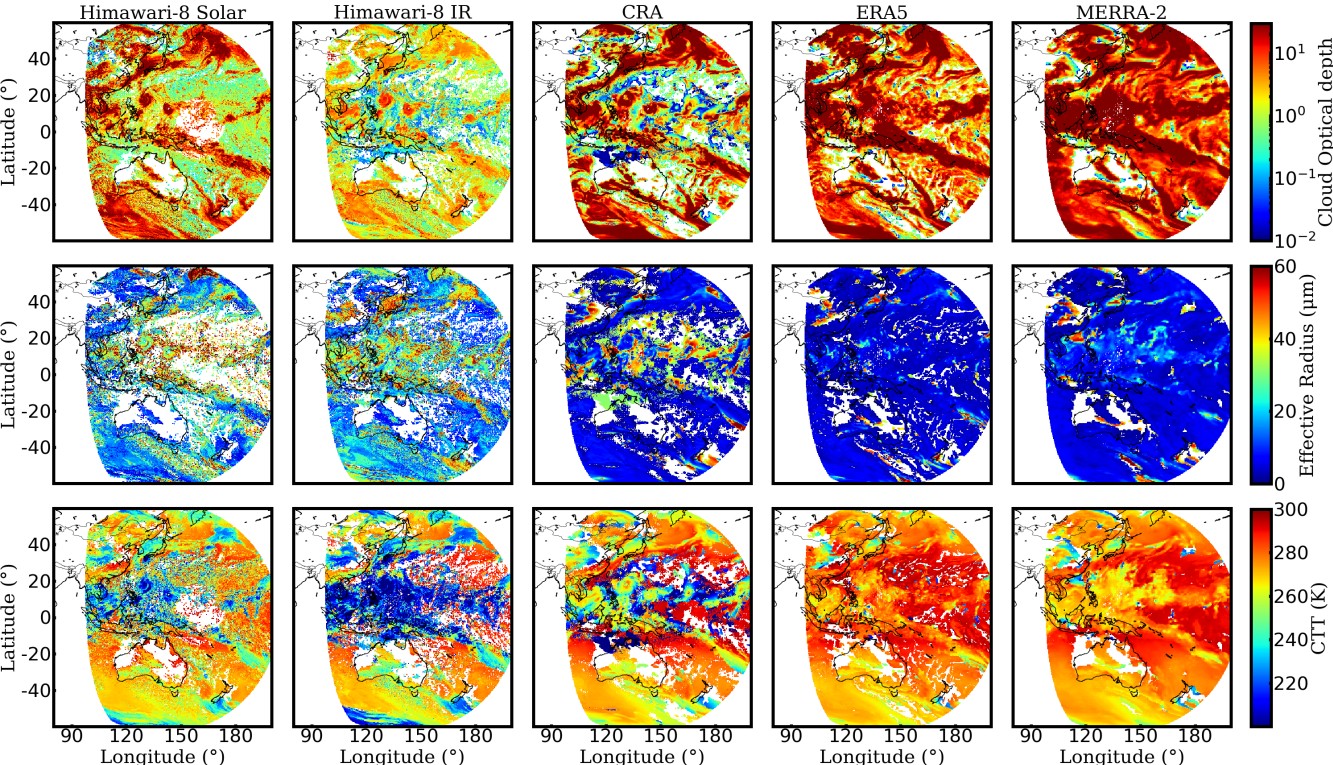

**Figure 1.** Comparison of cloud optical depth, effective radius, and cloud top temperature between the simulations and satellite retrieved cloud products. The results are taken at 00:00 (UTC) on 12 September 2016.



**Figure 2.** Joint histograms of cloud optical depth, effective radius, and cloud top temperature between satellite retrieved cloud products and simulations from the CRA reanalysis. The results are taken from 10 September 2016 to 17 September 2016, and more than one million pixels are included.

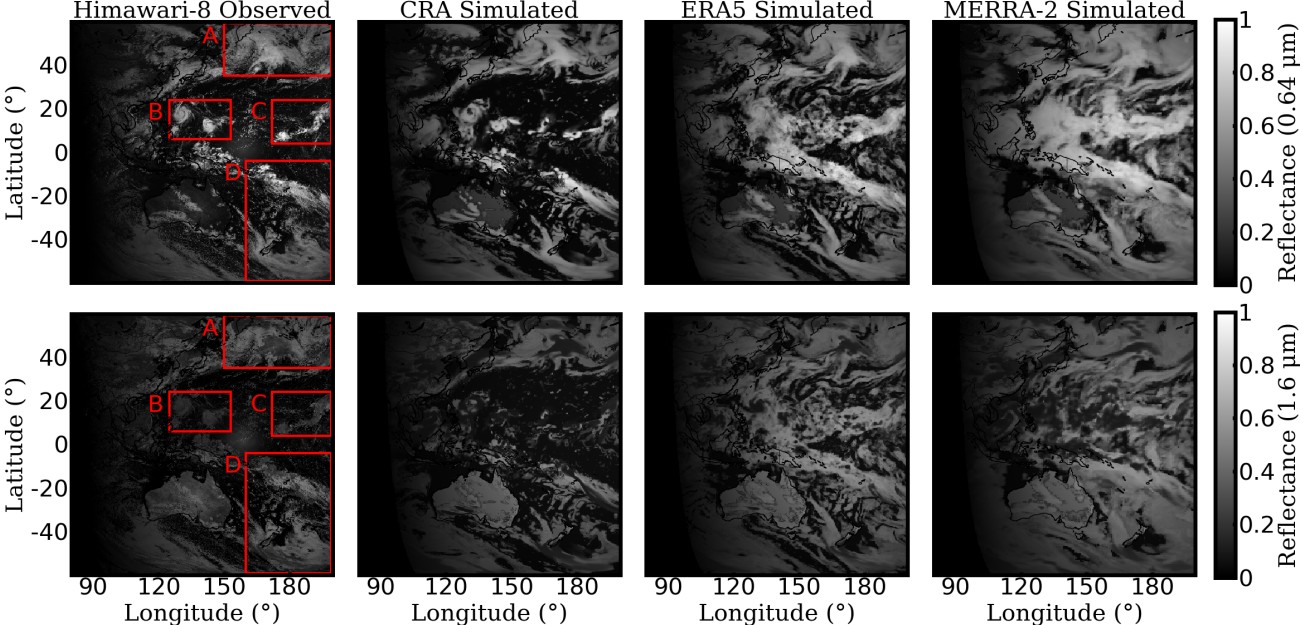

**Figure 3.** Observed and simulated reflectance in the 0.64-µm (top) and 1.6-µm (bottom) channels. The results are taken at
00:00 (UTC) on 12 September 2016.





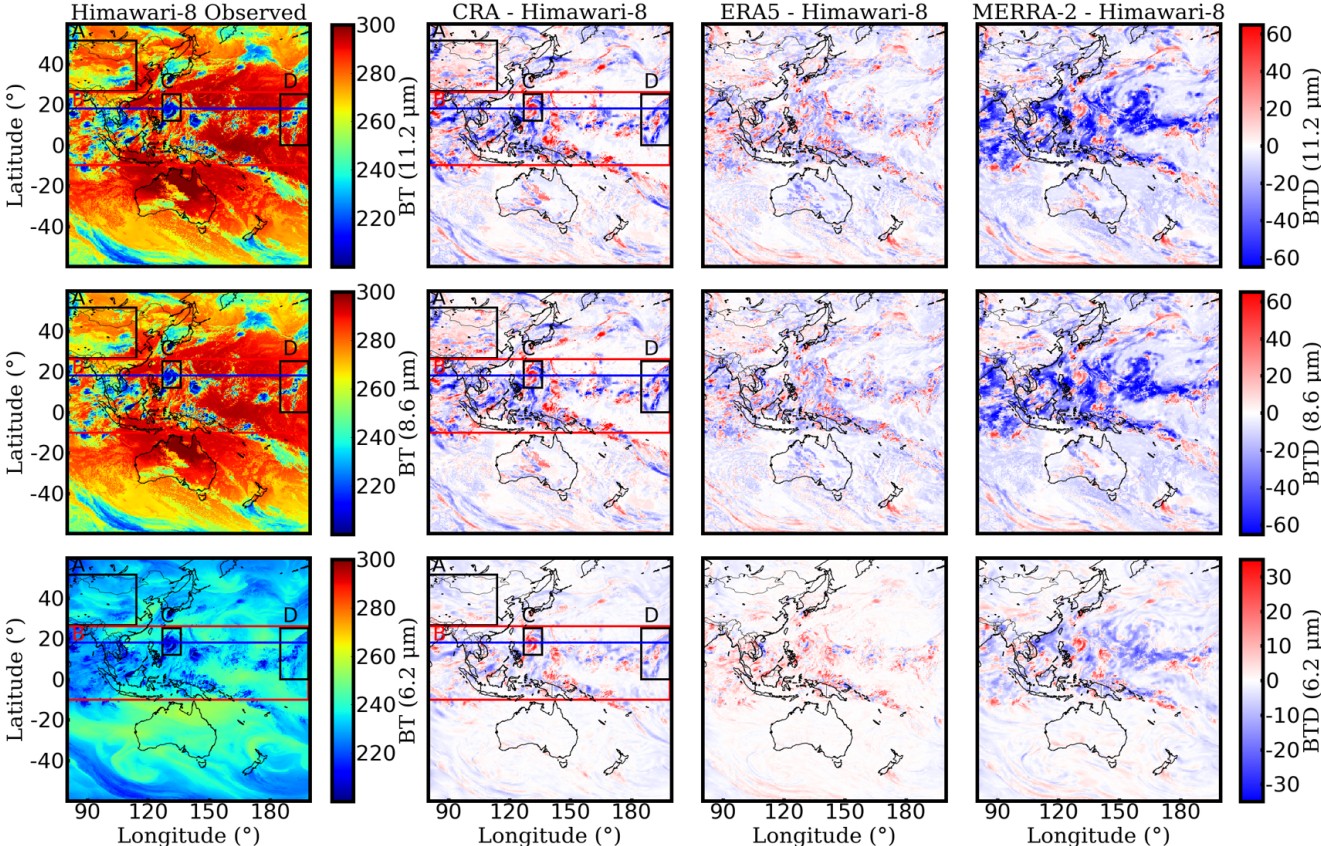

**Figure 4.** Observed results and the brightness temperature differences between the observations and simulations in the 11.2-μm (top), 8.6-μm (middle), and 6.2-μm (bottom) channels. The results are taken at the same time as that in Figure 3.


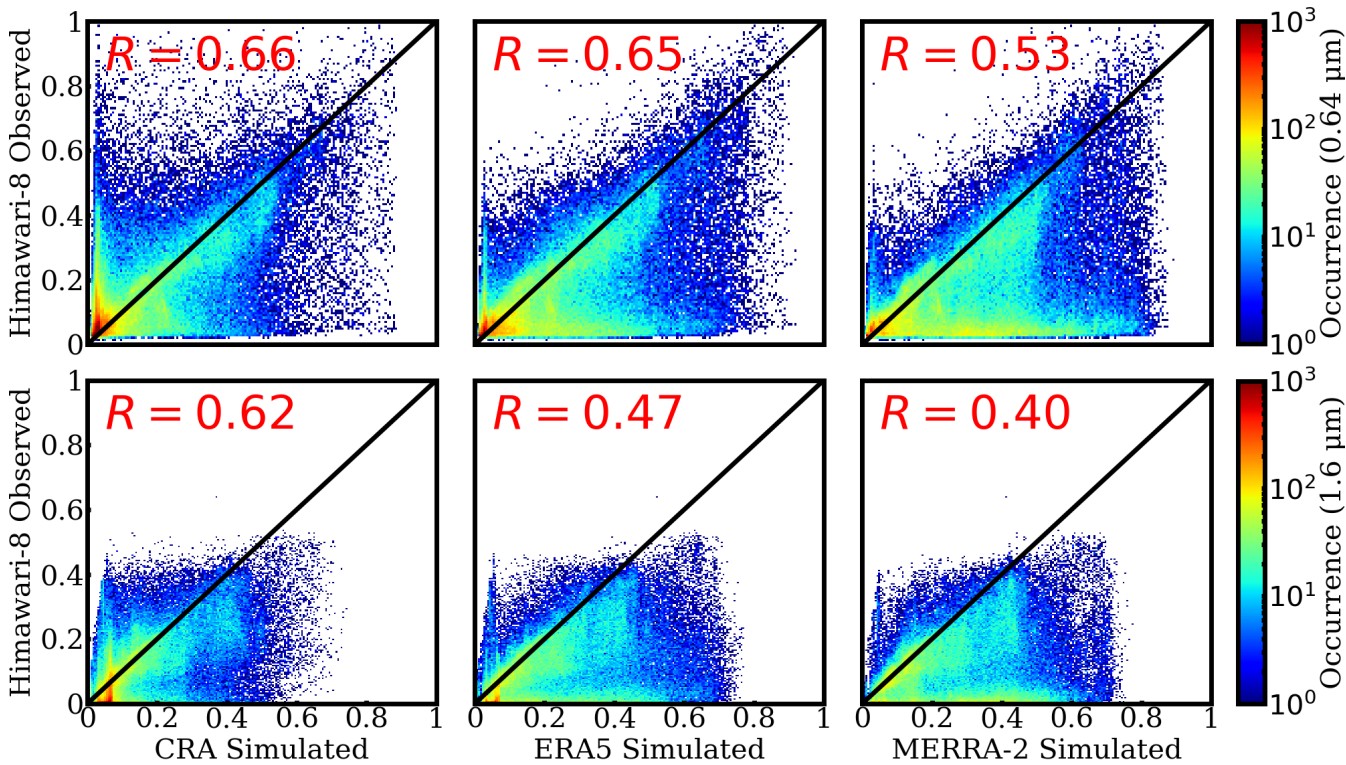

**Figure 5.** Pixel-to-pixel comparisons between the observed and simulated reflectance in the 0.64-μm (top) and 1.6-μm (bottom) channels. The histograms illustrate the occurrences of reflectance, and the results are taken at the same time as that in Figure 3.




**Figure 6.** Same as the results in Figure 5, but for the infrared channels.





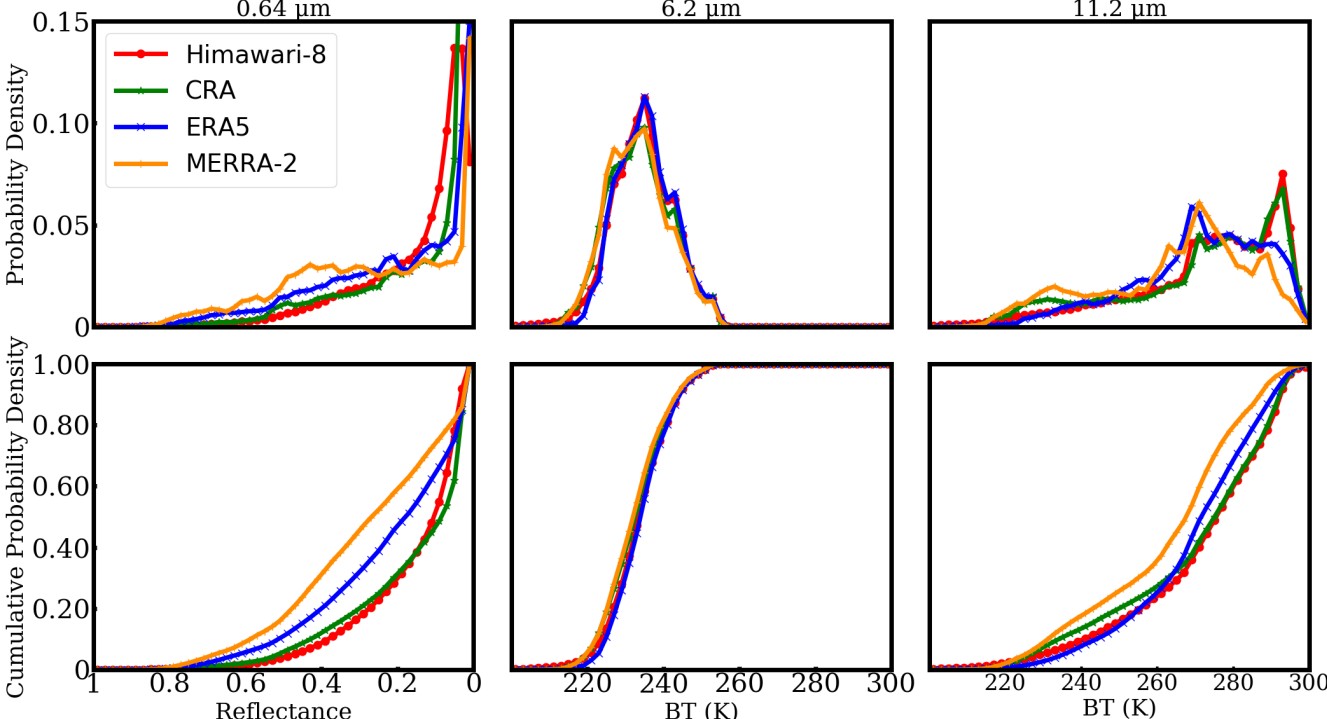

**Figure 7.** Probability and cumulative probability density for the observed and simulated results in the 0.64- (left), 6.2- (middle), and 11.2-µm (right) channels.

**Figure 8.** Comparison of the profiles of the temperature, cloud mixing ratio, cloud effective radius and optical depth in the
CRA (left), ERA5 (middle), and MERRA-2 (right) reanalysis datasets. The results are from Figure 4 marked by blue dashed
lines.





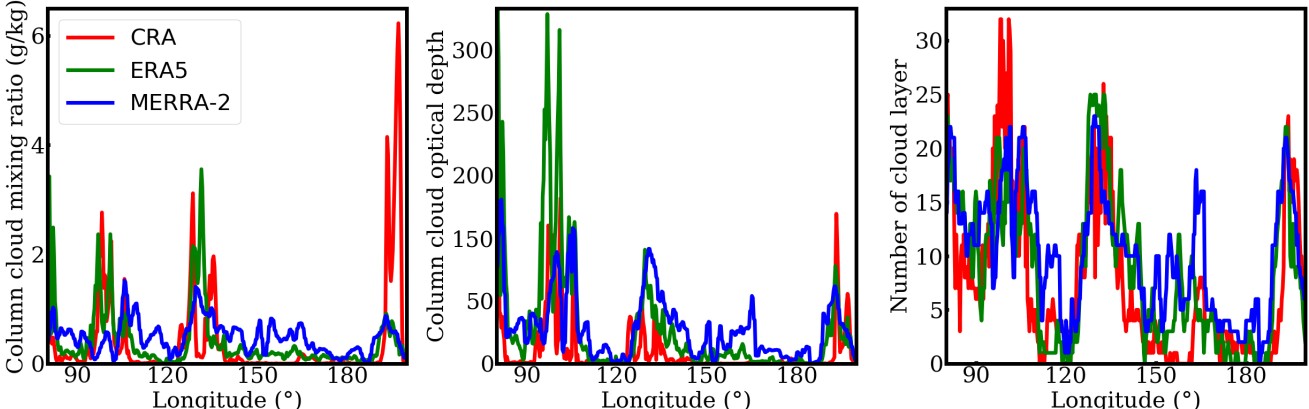

**Figure 9.** Column cloud mixing ratio (left), cloud optical depth (middle), and number of cloud layer (right) in each column.

The results are from Figure 4 marked by blue dashed lines.

**Figure 10.** Observed and simulated brightness temperature differences of 11.2–12.4-μm (top), 8.6–11.2-μm (middle), and 6.2–11.2-μm (bottom). The results are taken at the same time as that in Figure 3.




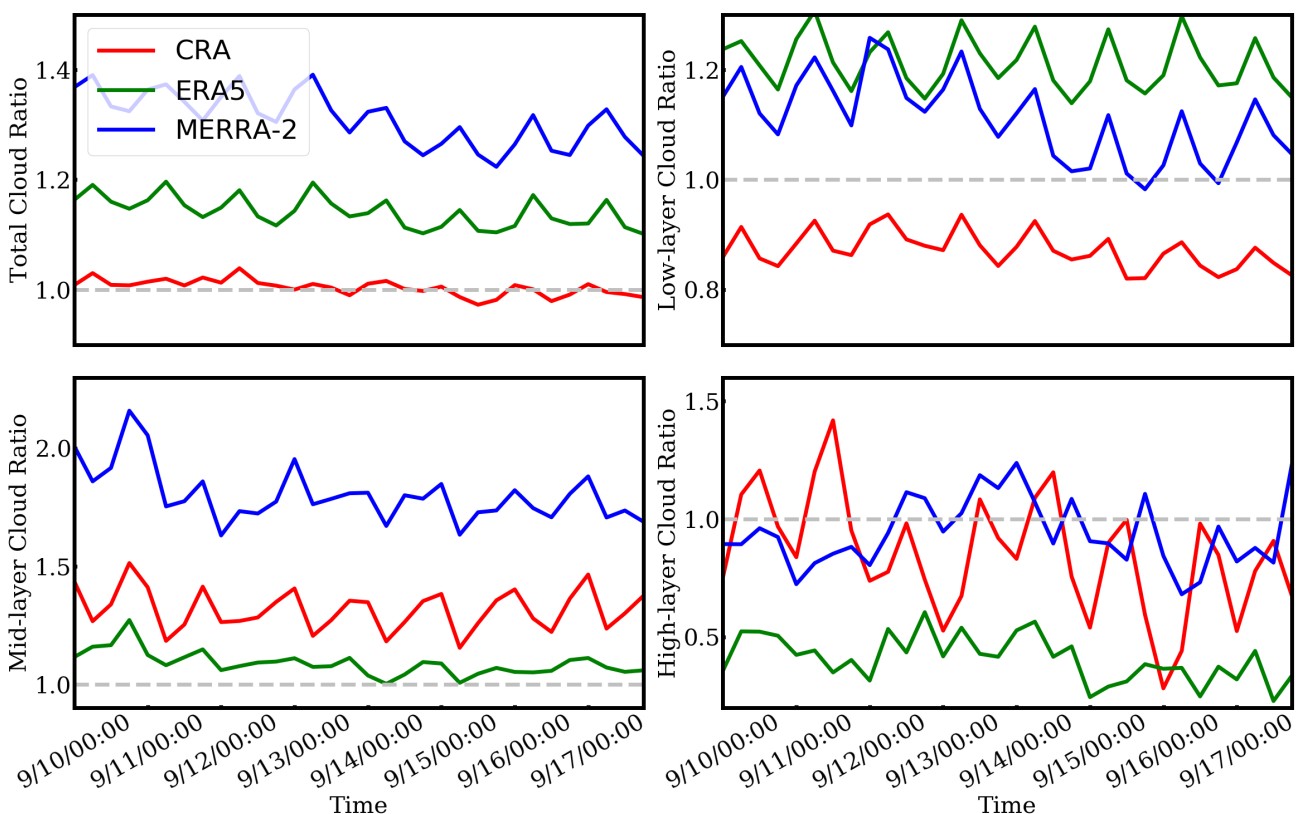

**Figure 11.** Temporal variation of the ratios (simulation-to-observation) for different layer clouds. The classification of clouds is based on the BTs in the 11.2-μm channel.





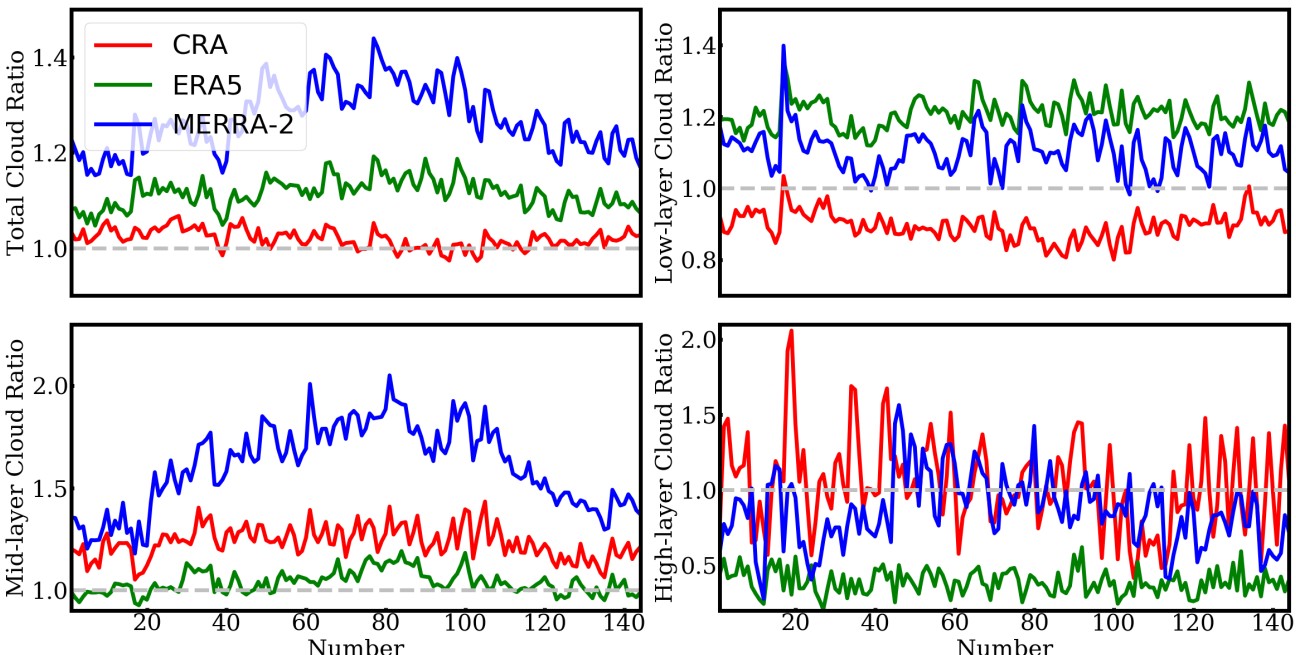


**Figure 12.** Same as Figure 11, but for the results from 144 realizations spanning over 2016.

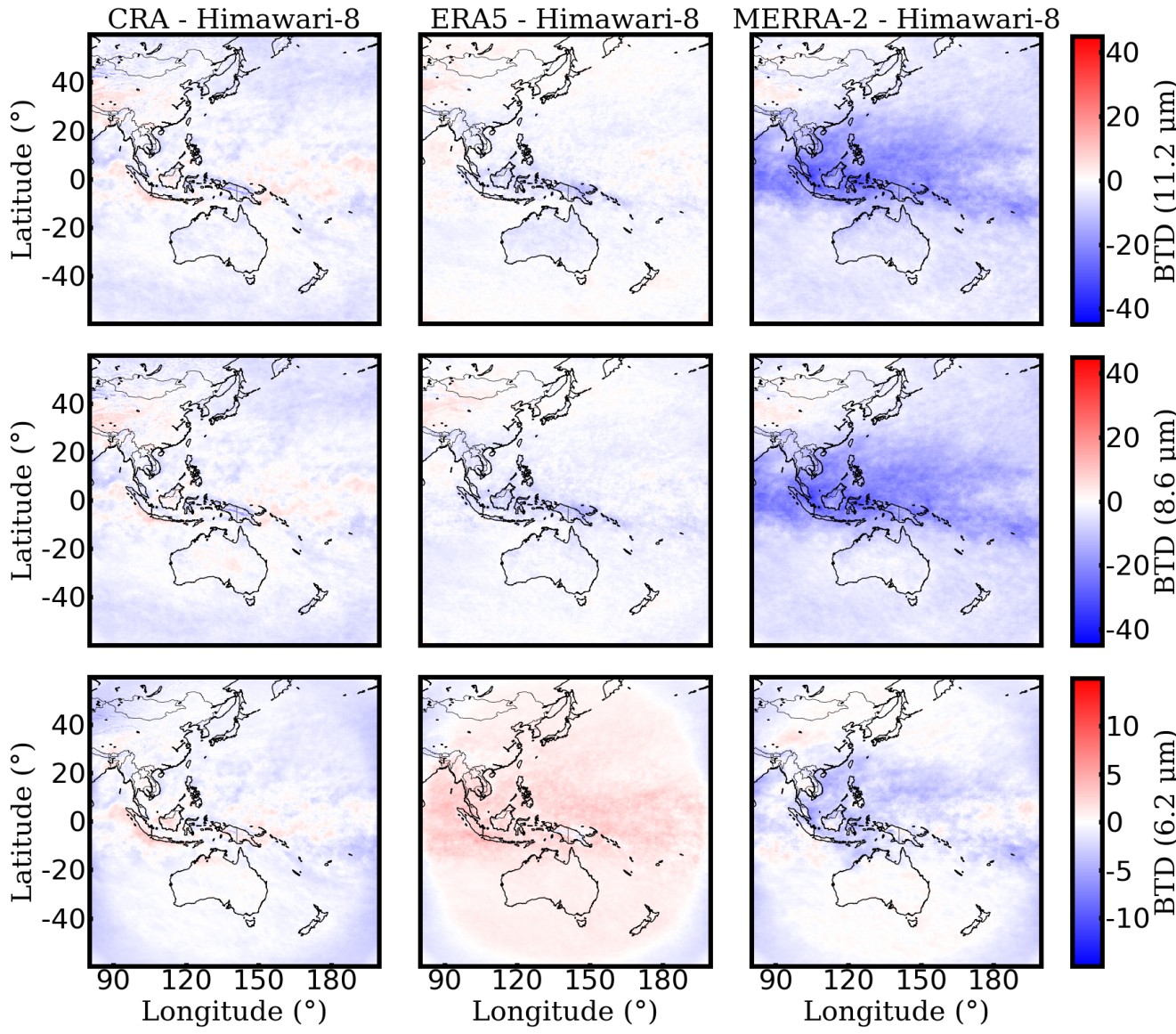

**Figure 13.** Average result of brightness temperature differences between the observations and simulations in the 11.2-μm (top), 8.6-μm (middle), and 6.2-μm (bottom) channels. The observations and simulations are from the 144 realizations spanning over 2016.




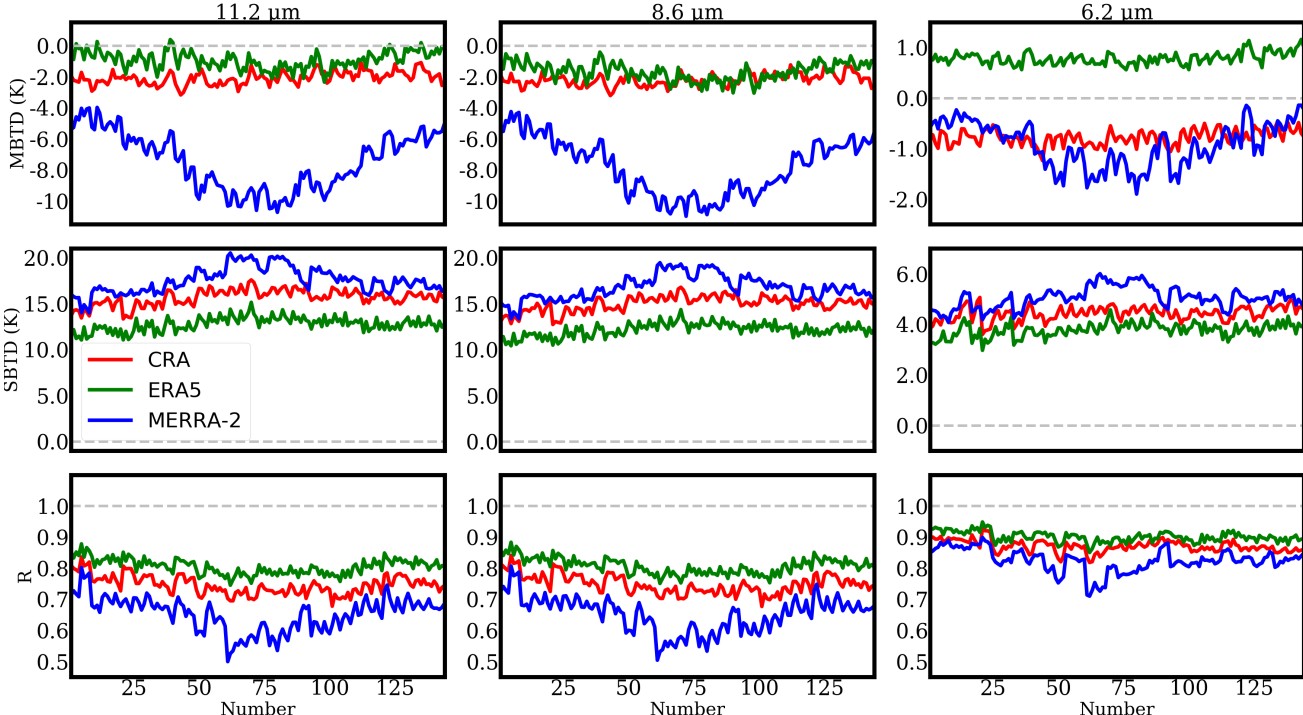

**Figure 14.** Temporal variation of three statistical parameters: the mean error (MBTD), standard error (SBTD), and correlation coefficient (R) between the observation and simulation. The results are from 144 realizations spanning over 2016.