# Peer review of "Assessment of cloud properties from the reanalysis with satellite observations over East Asia"

_Atmospheric Measurement Techniques, 2019_

## Referee Comment (RC1) · Anonymous Referee #2 · 30 Oct 2019

This paper by Yao et al., evaluates qualities of cloud properties in three reanalysis datasets, namely, China Meteorological Administration Reanalysis data (CRA), ECMWF's Fifth-generation Reanalysis (ERA5), and Modern-Era Retrospective Analysis for Applications version 2 (MERRA-2). A radiance-based evaluation approach is utilized with reflectance and brightness temperature observations from the Advanced Himawari Imager (AHI) onboard the Himawari-8 satellite. A radiative transfer model (CRTM) is used to link cloud related variables from reanalysis to satellite observations.

Overall, I believe this work is very valuable, which enhances our understanding of cloud representation in those reanalysis products. However, I have some concerns about the structure and some details of this paper.

Several major concerns I have about this paper include: 1. This paper uses observa-

tions from AHI/Himawari-8 to evaluate reanalysis. It is very important to mention that which satellite products (in particular cloud related datasets) are used as input in the three reanalysis products.

2. The advantages of a radiance-based evaluation approach are discussed in the abstract and introduction. I don't understand why the authors still use a lot of space describing AHI cloud products in Section 4?

3. This paper uses almost 4-pages to describe a case (a snapshot on a particular day) assessment, which I think is not necessary. In my point of view, the authors should pay more attention on long term cloud representation (e.g., cloud monthly mean, seasonal/annual variability).

Some minor suggestions include:

1. Page 2, large advantages of spatial distributions –> large advantages of spatial coverages

2. Page 6, CTT from two satellite retrieved cloud datasets (i.e., from solar and thermal infrared) How to use AHI solar bands to get CTT, can you give more details on this?

3. Figures 3, 5, and 7 The plots in Figures 5 and 7 use all pixels (i.e., clear + cloudy) in Figure 3? If yes, I suggest remove clear pixels or only focus on the regions of interest. I noticed that a large number of pixels in Australia are clear and reflectances from models are much higher (brighter) than AHI observations. This can significantly bias your plots in Figs. 5 and 7, and statistics.

4. Figures 11 and 12 and corresponding text: The authors use BT 11um as a proxy to differentiate clouds on low, mid, and high levels. This is problematic since high and thin cirrus may be attributed to low clouds.

---

## Referee Comment (RC2) · Anonymous Referee #3 · 22 Dec 2019

General Comments: This paper is about assessment of cloud properties from the reanalysis with satellite data over East Asia. Three sets of reanalysis data are used, including the newly developed China Meteorological Administration Reanalysis data (CRA), the ECMWF's Fifth-generation Reanalysis (ERA5), and the Modern-Era Retrospective Analysis for Applications, Version 2 (MERRA-2). And, to avoid the unrealistic assumptions and uncertainties on satellite retrieval algorithms and products, a radiative transfer model (CRTM) is used to transform reanalysis data into radiance/brightness temperature that can be directly compared with the Himawari-8 satellite data. Although cloud properties from CRA, ERA5, and MERRA-2 have their own advantages, the results show that ERA5 reanalysis data is best representative of cloudy atmosphere over East Asia, while the results in CRA are close to those in ERA5. This study may con-

tribute to the improvement of cloudy property representation in models and satellite observations. This paper is within the scope of Atmospheric Measurement Techniques but some improvement should be conducted before the paper could be accepted for publication.

Major concerns: 1. The authors claim that the radiance-based evaluation approach could avoid unrealistic assumptions and uncertainties on satellite retrieval algorithms and products, and thus it is a better way to carry out the assessment of cloud properties from various reanalysis. However, I would say I only partially agree with the authors on the perspective that the conventional way to compare cloud variables could be still indispensable. Without knowing the quantitative and qualitative differences in cloud properties, it is still hard to explain the radiance/brightness temperature differences resulting from the radiative transfer modeling. Thus, more discussion about the cloud optical properties should be added. 2. Previous studies (i.e., Yi et al., JGR, 2017a, b) indicate that a consistent cloud optical property parameterization scheme should be used in satellite retrievals and modeling studies to well simulate the radiance/flux at the top of the atmosphere under cloudy sky. Any mismatch in cloud optics parameterization could induce large bias in the retrieval and simulations. Taking that into account, it seems the study here using CRTM with a new set of cloud optical property look up tables (it is also not clear what kind of ice cloud particle model is used) that is inconsistent with the Himawari-8 cloud retrieval algorithm, could be potentially problematic in the satellite radiance/brightness temperature simulation. The authors may need to consider using the Voronoi ice scattering model by Letu et al. (2016; 2018). 3. Apart from the potential problem in cloud optical property, another important issue is about the differences in the atmospheric profiles. The simulated radiance/brightness temperature is closely related with the atmospheric profiles. Whereas, differences in the atmospheric profiles among the reanalysis datasets are prevalent. And these differences may contribute to the simulated results under cloudy sky. Thus, I think it would be best that the authors provide some analysis of the clear-sky evaluations (maybe in appendix). This would be helpful for the reader to distinguish the impacts of atmospheric profiles

and the cloud properties. 4. In part 3: methodology, to derive the necessary cloud property inputs for RTM, the authors also make quite a few assumptions. Especially in deriving the effective radius (Line 145), the used definition is somewhat different from those normally used in parameterization. As the effective radius is a very important quantity that decides the cloud optical properties in the parameterization, the authors need to analyze how the differences in the definition of effective radius will influence the results. 5. There are quite a few places in the text that are not clearly stated and are difficult to understand. For example: Line 301: It is not clear how the probability and cumulative probability are calculated here. And how do you "obviously" figure out from Figure 7 that "total cloud is overestimated in ERA5 and MERRA-2" ? Line 348: How do you define "ratio of the simulation-to-observation frequency of pixels with particular BTs"? Line 353: What does TCC mean? Line 376-377: How do you define mean error (MBTD) and standard error (SBTD) ? 6. Figure captions in this paper are not clear enough to show what the figures are about. For example, Figure 7 "Probability and cumulative probability density for the observed and simulated results ..." – what kind of "results" do you have here? The authors failed to state the name of the variable. Figure 8 "... The results are from Figure 4 marked by blue dashed lines" – couldn't see the "blue dash line" in Figure 4, and actually, there are too many elements in Figure 4.

Minor problems: Line 33: "The ERA5 reanalysis is found the most capability ..." should be "The ERA5 reanalysis is found to have the most capability ..." Line 97: Do you have some references for the CRA-interim? Line 142: "Ignore the uncertainties ..." should be "Ignoring the uncertainties ..."; In addition, is it reasonable to assume mixed phase cloud can be ignored? Line 187: "The correlation between the two is small." – This sentence is vague, as it is not clear about what are "the two". Line 191: "We notice that ..." should be "It is noted that ..." Line 215-217: The authors mentioned the cloud scattering properties in the CRTM are recalculated. Then some necessary validation and description are needed to prove the validity of the new implementation. Line 230: "From" should be "from" Line 272: "with a mean BTs of ..." should be "with a mean BT of ..." Line 324-325: "an abnormal excessive cloud mixing ratio" should be "an

abnormally excessive cloud mixing ratio" Line 373: "as marked in region A in Figure A" – where is Figure A? Line 390: "the in-site observation"? Line 413: "demonstrate that . . ." should be "demonstrating that . . ."

---

## Author Comment (AC1) · 6 Jan 2020

**Response to Reviewer # 1 (Manuscript ID: amt-2019-223)**

First of all, we would like to thank the reviewers for their valuable comments. In the revised manuscript, we have accommodated all the suggested changes into consideration and revised the manuscript accordingly. The reviewers' comments are copied here as texts in BLACK. The authors' responses are followed in BLUE, and our changes in the manuscript are in *italics*.

**Reviewer # 1**

This paper by Yao et al., evaluates qualities of cloud properties in three reanalysis datasets, namely,

China Meteorological Administration Reanalysis data (CRA), ECMWF's Fifth-generation Reanalysis (ERA5), and Modern-Era Retrospective Analysis for Applications version 2 (MERRA-2). A

radiance-based evaluation approach is utilized with reflectance and brightness temperature observations from the Advanced Himawari Imager (AHI) onboard the Himawari-8 satellite. A radiative transfer model (CRTM) is used to link cloud related variables from reanalysis to satellite observations.

Overall, I believe this work is very valuable, which enhances our understanding of cloud representation in those reanalysis products. However, I have some concerns about the structure and some details of this paper.

**Several major concerns I have about this paper include:**

1. This paper uses observations from AHI/Himawari-8 to evaluate reanalysis. It is very important to mention that which satellite products (in particular cloud related datasets) are used as input in the three reanalysis products.

**Response:** Thanks for the suggestion. Yes, it is necessary to introduce satellite observations assimilated for the reanalysis, because the differences on satellite datasets assimilated may be a potential reason for different performances of the reanalysis. Thus, we added the related contents in Section 2. Both ERA5

and CRA consider Himawari-8 observations, whereas MERRA-2 does not. This may be one of the reasons that MERRA-2 has relatively poor performance in the Asian region. To address the reviewer's concern, we included the following discussion in the revision (Lines 401-403):

"*It should be noticed that both ERA5 and CRA reanalysis consider Himawari-8 observations for*

*assimilation (see Section 2), whereas MERRA-2 dose not. This may be one of the reasons that*

*MERRA-2 has relatively poor performance on cloud representation in the Asian region.*"

35. The advantages of a radiance-based evaluation approach are discussed in the abstract and introduction. I don't understand why the authors still use a lot of space describing AHI cloud products in Section 4?

**Response:** In the original submission, we try to demonstrate more clearly that direct retrieval-based evaluation may be problematic, so Figures 1 and 2 as well as the corresponding discussions give comparisons based on the cloud products retrieved based on different bands (i.e., the solar channels and thermal infrared channels). We agree with the reviewer that the purpose of the study is to evaluate different reanalysis datasets based on the radiance-based approach. Considering that the Introduction Section is clear enough to demonstrate the disadvantage and uncertainties related to the retrieval-based evaluation (as noticed by the reviewer), we have removed the details related to the retrieval-based evaluation (i.e., Figs. 1 and 2 as well as the corresponding discussions), and the part related to AHI cloud products has also been removed.

47. This paper uses almost 4-pages to describe a case (a snapshot on a particular day) assessment, which I think is not necessary. In my point of view, the authors should pay more attention on long-term cloud representation (e.g., cloud monthly mean, seasonal/annual variability).

**Response:** Actually, the "case study" mentioned in this study is not a snapshot for a particular day, and we consider results over eight days with over 30 realizations. To avoid such misunderstanding, we have the added the following sentence in the revision:

*"Noted that even for this case study, we consider a period over eight days covering 32 time steps."*

We think the case assessment is meaningful as well for the following reasons:

(1). The results in Figures 11 and 13 indicate that the evaluations are generally stable over time. The results of the case study are universalistic and representative, and the corresponding conclusions are actually consistent with those from the long-term evaluation. However, because the forward radiative transfer simulation is computationally expensive, this study considers results from a typical case with eight days and a generally evaluation with 144 realizations over one year.

(2). In fact, we use the case study results to present more details of the three reanalysis, whereas use the long-term results for the general evaluation. As a result, we think both parts are necessary.

(3). Both the case study and the 144 realizations spanning over one year indicate that our methodology, i.e., the radiance-based evaluation, is feasible, and the results are reliable.

Meanwhile, we agree with the reviewer that more attentions should also be paid to cloud monthly mean, seasonal/annual variability, and we have extended these discussions. Furthermore, we would like to investigate the long-term cloud representation in details in our future studies.

**Some minor suggestions include:**

1.  Page 2, large advantages of spatial distributions –> large advantages of spatial coverages.

**Response:** Thanks for your suggestion, and the phrase is corrected.

2.   Page 6, CTT from two satellite retrieved cloud datasets (i.e., from solar and thermal infrared) How to use AHI solar bands to get CTT, can you give more details on this?

**Response:** Sorry for the confusion because of my incorrect description. The cloud top in the product from Letu et al. (2018) is retrieved based on the observations in the infrared window channel (11.2 μm), and the cloud product of Iwabuchi et al. (2018) is based on observations in the 10.4 μm channel.

However, the atmospheric profiles used in the cloud retrieval are different, and Letu et al. (2018) and

Iwabuchi et al. (2018) cloud products use profiles from the GPV (the Grid Point Values of atmospheric)

and MERRA reanalysis, respectively. As mentioned above, we think this study should focus on the radiance-based evaluation, so we have removed the section on cloud retrieval products.

3.   Figures 3, 5, and 7. The plots in Figures 5 and 7 use all pixels (i.e., clear + cloudy) in Figure 3? If yes, I suggest remove clear pixels or only focus on the regions of interest. I noticed that a large number of pixels in Australia are clear and reflectances from models are much higher (brighter) than AHI

observations. This can significantly bias your plots in Figs. 5 and 7, and statistics.

**Response:** Yes, both clear and cloudy pixels are considered in Figs 5 and 7. Because we consider different clouds by using different BTs or BTDs, even with all pixels considered, the problems related to the reanalysis over cloudy regions can be illustrated by the figures. We think the reviewer gives an excellent comment to consider only cloudy pixels, so we added a new Figure 5 in the revision with clear and cloudy pixels considered separately. We found that the cloud property representation contributes more to the differences than the atmospheric profiles.

Meanwhile, as there is no "truth" for the classification of clear/cloudy pixels (again, we do not want to use the retrieval results due to their own uncertainties), we can only use reanalysis data for the classification. This is also a reason that we mostly consider all pixels in the discussions.

4.   Figures 11 and 12 and corresponding text: The authors use BT 11um as a proxy to differentiate clouds on low, mid, and high levels. This is problematic since high and thin cirrus may be attributed to low clouds.

**Response:** Thanks for the suggestion. In the revision, the widely-used thresholds based on BTDs between the 6.2- and 11.2-μm channels are used to differentiate clouds over different layers (Mecikalski and Bedka, 2006; Yao et al., 2018). Because of strong water vapor absorption in the 6.2-μm channel and the temperature lapse rate within the troposphere, the BTDs between 6.2- and 11.2-μm are usually negative. The BTDs increase as the cloud top height increases and larger negative BTDs often corresponds to clear-sky pixels. We use the thresholds of -45 to -30 K to infer pixels with low cloud tops, and those with low- to mid-layer cloud are represented by BTDs between -30 and -10 K following

Mecikalski and Bedka (2006). The BTDs less than -45 K normally correspond to clear pixels and those larger than -10 K are from high cloud pixels. With the improved classification, most results and conclusion are similar, and slight differences are noticed for mid-layer clouds (The mid-layer cloud in

CRA is closest to the observation.) Thanks for your suggestions, and we have updated the corresponding classification, figures, and the corresponding discussion in the revision.

**References:**

Letu, H., Nagao, T. M., Nakajima, T. Y., Riedi, J., Ishimoto, H., et al.: Ice cloud properties from

Himawari-8/AHI next-generation geostationary satellite: capability of the AHI to monitor the DC

cloud generation process, IEEE Trans. Geosci. Remote Sens. 12, 1-11, 2018.

Iwabuchi, H., Putri, N. S., Saito, M., Toloro, Y., Sekiguchi, M., et al.: Cloud property retrieval from multiband infrared measurements by Himawari-8, J. Meteor. Soc. Jpn, 96, 27-42, 2018.

Mecikalski, J. R. and Bedka, K. M.: Forecasting convective initiation by monitoring the evolution of moving cumulus in daytime GOES imagery, Mon. Wea. Rev., 134, 49-78, 2006.

Yao, B., Liu, C., Yin, Y., Zhang, P., Min, M., and Han, W.:Radiance-based evaluationo WRF cloud properties over East Asia:Direct comparison with FY-2E observations, J. Geophys. Res., 123,

4613-4629, 2018.

---

## Author Comment (AC2) · 6 Jan 2020

**Response to Reviewer # 2 (Manuscript ID: amt-2019-223)**

First of all, we would like to thank the reviewers for their valuable comments. In the revised manuscript, we have accommodated all the suggested changes into consideration and revised the manuscript accordingly. The reviewers' comments are copied here as texts in BLACK. The authors' responses are followed in BLUE, and our changes in the manuscript are in *italics*.

**Reviewer # 2**

General Comments: This paper is about assessment of cloud properties from the re- analysis with satellite data over East Asia. Three sets of reanalysis data are used, including the newly developed

China Meteorological Administration Reanalysis data (CRA), the ECMWF's Fifth-generation

Reanalysis (ERA5), and the Modern-Era Retrospective Analysis for Applications, Version 2

(MERRA-2). And, to avoid the unrealistic assumptions and uncertainties on satellite retrieval algorithms and products, a radiative transfer model (CRTM) is used to transform reanalysis data into radiance/brightness temperature that can be directly compared with the Himawari-8 satellite data.

Although cloud properties from CRA, ERA5, and MERRA-2 have their own advantages, the results show that ERA5 reanalysis data is best representative of cloudy atmosphere over East Asia, while the results in CRA are close to those in ERA5. This study may contribute to the improvement of cloudy property representation in models and satellite observations. This paper is within the scope of

Atmospheric Measurement Techniques but some improvement should be conducted before the paper could be accepted for publication.

**Major concerns:**

1. The authors claim that the radiance-based evaluation approach could avoid unrealistic assumptions and uncertainties on satellite retrieval algorithms and products, and thus it is a better way to carry out the assessment of cloud properties from various reanalysis. However, I would say I only partially agree with the authors on the perspective that the conventional way to compare cloud variables could be still indispensable. Without knowing the quantitative and qualitative differences in cloud properties, it is still hard to explain the radiance/brightness temperature differences resulting from the radiative transfer modeling. Thus, more discussion about the cloud optical properties should be added.

**Response:** We agree with the reviewer that the comparisons with retrieved cloud products are still necessary for assessment of model simulations. As we have discussed in the Introduction Section (as well as Figs. 1 and 2 in the original submission), such direct comparison may be also problematic due to the uncertainties related to retrieval product. Of course, the radiance-based evaluation has its own disadvantages as well. Thus, we decided to focus only on the radiance-based evaluation, and more detailed quantitative and qualitative evaluation based on direct comparison is suggested be performed in further independent studies. Besides removing the retrieval-based evaluation parts, we also included the following discussion in the revision (Lines 73-77):

*"The retrieval-based evaluation is an indispensable approach in the evaluation of atmospheric properties from various simulations, and quantitative and qualitative analysis of the cloud optical properties, e.g., the cloud effective radius and optical depth, can be evaluated directly. However, to avoid uncertainties associated with satellite retrieval algorithms and platforms, another alternative radiance-based comparison is chosen for the cloud properties assessment in our study."*

2. Previous studies (i.e., Yi et al., JGR, 2017a, b) indicate that a consistent cloud optical property parameterization scheme should be used in satellite retrievals and modeling studies to well simulate the radiance/flux at the top of the atmosphere under cloudy sky. Any mismatch in cloud optics parameterization could induce large bias in the retrieval and simulations. Taking that into account, it seems the study here using CRTM with a new set of cloud optical property look up tables (it is also not clear what kind of ice cloud particle model is used) that is inconsistent with the Himawari-8 cloud retrieval algorithm, could be potentially problematic in the satellite radiance/brightness temperature simulation. The authors may need to consider using the Voronoi ice scattering model by Letu et al. (2016; 2018).

**Response:** We agree with the reviewer that inconsistent cloud optical property models could be a potential problem for the differences in different satellite retrievals. This is the reason that we think the retrieval-based evaluation can be problematic. We have omitted the figures showing the direct comparison. In our radiance-based evaluation, no satellite cloud product is used, so such differences for different cloud product will not influence our results.

Meanwhile, we clarified that the optical properties of aggregate columns with eight elements and severe surface roughness are used for CRTM. We think it is interesting to check the influence of cloud optical property parameterization on our evaluation, and this is suggested as a future study as following (Lines 173-176):

*"It should be noted that schemes for both cloud optical properties (e.g., ice cloud model) in the RTM and coupling between atmospheric reanalysis and RTM (e.g., approximation of cloud effective radius) may influence simulated BTs/reflectances, although the influences are relatively minor compared to presences of clouds (cloud amount). The potential numerical uncertainties due to different schemes will be performed with more details in further studies."*

3. Apart from the potential problem in cloud optical property, another important issue is about the differences in the atmospheric profiles. The simulated radiance/brightness temperature is closely related with the atmospheric profiles. Whereas, differences in the atmospheric profiles among the reanalysis datasets are prevalent. And these differences may contribute to the simulated results under cloudy sky. Thus, I think it would be best that the authors provide some analysis of the clear-sky evaluations (maybe in appendix). This would be helpful for the reader to distinguish the impacts of atmospheric profiles and the cloud properties.

**Response:** Thanks for the suggestion. It is interesting and meaningful to consider the cloudy and clear-sky pixels separately and to evaluation the contributions from cloud or atmospheric profiles.

(1) First, for the solar channel results (Figs. 1 and 3 in the new version), the differences are almost all contributed by cloud representation, because atmospheric profiles have little effect on the reflectance in the 0.64- and 1.6-$\mu$m channel. We added brief discussion and analysis in the revised paper.

(2) Comparison between simulated and observed BTs in the IR channels does show the overall performances of the reanalysis data due to both cloudy and atmospheric profiles. However, the discussion and classification based on BTDs can significantly remove the influence of atmospheric profiles, because the BTDs between the selected channels are mostly influenced by the cloud properties (e.g., cloud height and cloud amount).

(3) Furthermore, we include the following discussions in the revision. If pixels are separated as cloudy or clear ones based on a criterion of 0.1 for the integrated column cloud optical depth in each pixel, the figure below shows the pixel-to-pixel comparisons between observed and simulated BTs in the 11.2-$\mu$m channel. The top row is for cloudy pixels, and the bottom one is for clear-sky pixels. Larger correlation values for the clear pixels indicate that the cloud properties do significantly contribute to the differences.

[Figure]

**Figure 1.** Pixel-to-pixel comparisons between the observed and simulated BTs in the 11.2-μm channel.

Top panels indicate the comparison for cloudy pixels, and the bottom panels show the comparison for clear pixels. The results are taken at 00:00 (UTC) on 12 September 2016.

(4)  Last but not the least, the reviewer raised an interesting and important point, which should and will be done in the future, we have added the following discussion (Lines 410-413):

"*The radiance-based approach is a reliable choice for the evaluation to avoid uncertainties due to*

*retrieval products, and its drawbacks may be investigated in further studies. For examples, differences*

*between simulated and observed radiances can be contributed by both cloudy and atmospheric*

*variables, and these may be distinguished by considering the same atmospheric profiles in the RTM*

*simulations.*"

4. In part 3: methodology, to derive the necessary cloud property inputs for RTM, the authors also make quite a few assumptions. Especially in deriving the effective radius (Line 145), the used definition is somewhat different from those normally used in parameterization. As the effective radius is a very important quantity that decides the cloud optical properties in the parameterization, the authors need to analyze how the differences in the definition of effective radius will influence the results.

**Response:** The reviewer noticed an important point of our study. In fact, the couple between reanalysis cloud variables and RT simulations is one of the most essential parts of this study. We have tried our best to avoid empirical relationships for cloud property estimation.

(1) For water cloud, the effective radius scheme is based on Thompson et al. (2004) a popular scheme
in mesoscale meteorological forecast models (e.g., the WRF model). The cloud number
concentration over continent and ocean regions are assumed as typical and widely used values
(Miles et al. 2000; Thompson et al., 2004; Wendisch and Yang, 2012).

(2) For ice clouds, the effective radius is physically estimated by mass extinction coefficient, which is
given by an empirical relationship related to ice water content (Heymsfield and McFarquhar, 1996;
Platt, 1997; Heymsfield et al. 2003), and the ice water content is from reanalysis directly.

(3) As also noticed by the reviewer, the coupling is far from being a done work. There could be
multiple ways to estimate the effective radius. For example, in our previous study (Yao et al. 2018),
the effective radius of ice particle is calculated based on ice crystal mass and mass-radius relation
(Hong et al. 2004). The following table compares observations with simulated BTs calculated
based on the schemes used in this study (Scheme A) and the previous study (Scheme B, Yao et al.
2018). The correlations between observations and simulations from two different radius
parameterized schemes are close to each other, and slight differences are noticed for the mean BT
differences (MBTD) and BTD standard deviation (SBTD). This indicates that the schemes for
effective radius estimation matter, whereas the influences are limited. Considering the length and
focus of this study, we will not include such discussion in the manuscript, but we do think such
sensitive study is interesting for a further study.

**Table 1.** The mean BT difference (MBTD), BTD standard deviation (SBTD), and correlation
coefficient (R) between the observation and simulations (simulations based on two different particle
effective radius estimations).

| Varibales | 6.2-μm | | 11.2-μm | |
|---|---|---|---|---|
| | Scheme A | Scheme B | Scheme A | Scheme B |
| R | 0.87 | 0.85 | 0.70 | 0.68 |
| MBTD (K) | -0.52 | -1.71 | -1.71 | -6.43 |
| SBTD (K) | 4.98 | 4.98 | 16.13 | 18.50 |

5. There are quite a few places in the text that are not clearly stated and are difficult to understand. For
example:

Line 301: It is not clear how the probability and cumulative probability are calculated here. And how do you "obviously" figure out from Figure 7 that "total cloud is overestimated in ERA5 and MERRA-2"?

**Response:** Here the probability and cumulative probability indicate the occurrence of pixels with certain BTs.

The probability ($P_{BT_o}$) is numerically calculated as:

$$P_{BT_o} = \frac{Number\ of\ pixels\ with\ BT\ between\ BT_o - \Delta BT\ and\ BT_o + \Delta BT}{Total\ pixel\ number}$$

, and the cumulative probability ($C_{BT_o}$) is given by:

$$C_{BT_o} = \frac{Number\ of\ pixels\ with\ BT\ less\ than\ BT_o}{Total\ pixel\ number}$$

The cumulative probability distribution is a good metric to give the occurrence of cloud. If we simply use a BT threshold of ~ 275K in the 11.2-μm channel to distinguish the cloud (BT < the threshold) and clear-sky (BT > the threshold) pixels, the cumulative probability with BTs less than 275K is approximate 0.8 and 0.7 for MERRA-2 and ERA5, respectively, whereas the cumulative probability with BTs less than 270-280 K for CRA and Himawari-8 observation is only 0.6. This suggests that over the observational domain, ~80% of the MERRA-2 and ~70% of the ERA are covered by clouds, which is larger than that from the observation.

We have rephrased the discussion and analysis in the corresponding paragraph.

Line 348: How do you define "ratio of the simulation-to-observation frequency of pixels with particular

BTs"?

**Response:** The "ratio of the simulation-to-observation frequency of pixels with particular BTs" is defined by the ratio of number of pixels with particular BT interval in simulation and observation. The value (RA) is numerically given by:

$$RA = \frac{Number\ of\ simulated\ pixles\ with\ between\ BT_a\ and\ BT_b}{Nummber\ of\ observed\ pixles\ with\ between\ BT_a\ and\ BT_b}$$

To better distinguish different clouds, the threshold of BTDs of 6.2 – 11.2-μm is used in the revision, and the corresponding explanation and discussion in the paragraph are rephrased.

Line 353: What does TCC mean?

**Response:** TCC here is the abbreviation of Total Cloud Cover, we have add the full name of it.

Line 376-377: How do you define mean error (MBTD) and standard error (SBTD) ?

**Response:** For each snapshot, the MBTD is the mean BTDs over the entire comparing region, and the

SBTD is the corresponding standard deviation. The MBTD and SBTD are calculated over the whole

Himawari-8 observation domain between simulated and observed BTs. We have clarified this in the revision.

6. Figure captions in this paper are not clear enough to show what the figures are about. For example:

Figure 7 "Probability and cumulative probability density for the observed and simulated results . . ." –

what kind of "results" do you have here? The authors failed to state the name of the variable.

**Response:** Sorry for the confusion. The "results" means the observed and simulated BTs or reflectances.

We have rephrased the captions.

Figure 8 " The results are from Figure 4 marked by blue dashed lines" – couldn't see the "blue dash line"

in Figure 4, and actually, there are too many elements in Figure 4.

**Response:** Sorry for the mistake. The caption has been changed into "*The profiles are for the track*

*marked by blue solid lines Figure 2.*". The regions or tracks particular discussed in the text are marked by boxes or lines in the new Figure 2, and we have improved the figure. Furthermore, to present Figure

8 more clearly, we have removed the cloud mixing ratio panels.

**Minor problems:**

Line 33: "The ERA5 reanalysis is found the most capability . . ." should be "The ERA5 reanalysis is found to have the most capability . . ."

**Response:** Thanks, and we have updated the sentence.

Line 97: Do you have some references for the CRA-interim?

**Response:** Because the CRA reanalysis dataset is producing and it will be released in 2020, and only a few papers have been published. Two papers by Liao et al. (2018) and Wang et al. (2018), which discuss the datasets assimilated in the CRA, have been referred in the revision.

Line 142: "Ignore the uncertainties . . ." should be "Ignoring the uncertainties . . ."; In addition, is it reasonable to assume mixed phase cloud can be ignored?

**Response:** Thanks. We have changed the "Ignore the uncertainties …" to "Ignoring the uncertainties".

In our study, we distinguish cloud with different phases based on the temperature profiles, so the mixed clouds are treated ice cloud and they are not ignored. We have tested that this would lead little bias, and clarified this in the revision.

Line 187: "The correlation between the two is small." – This sentence is vague, as it is not clear about what are "the two".

**Response:** It should be "the correlation between the CTT from CRA and the CTT from satellite retrieval based on the solar measurement". The section has been removed in the revision.

Line 191: "We notice that . . ." should be "It is noted that . . ."

**Response:** Thanks and we have removed the paragraph.

Line 215-217: The authors mentioned the cloud scattering properties in the CRTM are recalculated.

Then some necessary validation and description are needed to prove the validity of the new implementation.

**Response:** The validation of the CRTM was done in our previous study (Yao et al., 2018). As discussed in Figure 1 of Yao et al. (2018), the BTDs between the CRTM and rigorous (DISROT+LBLRTM)

simulations for ice and water clouds in different channels are generally less than 1 K, and they coverage to 0 K as cloud optical thickness increases to 10 or larger. We have added some discussion on the validation of the cloud optical properties in the CRTM model in the revision.

Line 230: "From" should be "from"

**Response:** Corrected.

Line 272: "with a mean BTs of . . ." should be "with a mean BT of ..."

**Response:** Thanks, and it has been corrected.

Line 324-325: "an abnormal excessive cloud mixing ratio" should be "an abnormally excessive cloud mixing ratio"

**Response:** Corrected.

Line 373: "as marked in region A in Figure A" – where is Figure A?

**Response:** It should be Figure 2 in the revision, and we have changed it.

Line 390: "the in-site observation"?

**Response:** We have changed it into "the in-situ observation"

Line 413: "demonstrate that . . ." should be "demonstrating that . . ."

**Response:** Thanks, and it has been corrected.

**References:**

Miles, N. Y., Verlinde, J., and Clothiaux, E. E.: Cloud Droplet Size Distributions in Low-Level Stratiform Clouds, J. Atmos. Sci., 57, 295-311, 2000.

Thompson, G., Rasmussen, R. M., and Manning, K.: Explicit forecasts of winter precipitation using an improved bulk microphysics scheme, Part I: Description and sensitivity analysis, Mon. Wea. Rev., 132, 519-542, 2004.

Wendish, M., and Yang, P.: Theory of Atmospheric Radiative Transfer. A Comprehensive Introduction, 1$^{st}$ ed., WILEY-VCH Verlag Gmbh, Weinheim, Germany, 2012.

Heymsfield, A. J., and McFarquhar, G. M.: On the high albedos of anvil cirrus in the tropical Pacific warm pool: Microphysical interpretations from CEPEX and from Kwajalein, Marshall Islands, J. Atmos. Sci., 53, 2424-2451, 1996.

Platt, C. W. R.: A Parameterization of the Visible Extinction Coefficient of Ice Clouds in Terms of the Ice/Water Content, J. Atmos. Sci., 54, 2083-2098, 1997.

Heymsfield, A. J., Matrosov, J. S., and Baum, B.: Ice water path optical depth relationships for cirrus and deep stratiform ice cloud layers, J. Appl. Meteor., 42, 1369-1390, 2003.

Yao, B., Liu, C., Yin, Y., Zhang, P., Min, M., and Han, W.: Radiance-based evaluation o WRF cloud properties over East Asia:Direct comparison with FY-2E observations, J. Geophys. Res., 123, 4613-4629, 2018.

Liao, J., Hu, K., Jiang, H., Cao, J., Jiang, L., Li, Q., Zhou, Z., Liu, Z., Zhang, T., and Wang, H.: Pre-Process and Data Selection for Assimilation of Conventional Observations in the CMA Global Atmospheric Reanalysis, Advances in Met S&T., 8, 133-142, 2018.

Wang, M., Yao, S., Jiang, L., Liu, Z., Shi, C., Hu, K., Zhang, T., Zhang, Z., and Liu, J.: Collection and Pre-Processing of Satellite Remote-Sensing Data in CRA-40 (CMA's Global Atmospheric ReAnalysis), Advances in Met S&T., 8, 158-163, 2018.

---

## Author Response (AR2)

**Response to the reviewer**

Thank the reviewer for his/her review and valuable comments. In the revised manuscript, we have accommodated the suggested changes into consideration and revised the manuscript accordingly. The reviewers' comments are copied here as texts in BLACK. The authors' responses are followed in BLUE.

Review Comments of "Assessment of cloud properties from the reanalysis with satellite observations over East Asia" by Yao et al.

**General Comments:**

I think after the first round of review and revision, the manuscript is greatly improved and I believe it is close to be ready for publication. The authors mostly addressed my questions well and I have no further comments on those questions. But I still have a few more comments that I think will be useful for the manuscript improvements.

1. I think the title of the manuscript can be revised to illustrate the highlight of this study by including "radiance-based approach" term in the title, for example, "Assessment of cloud properties from the reanalysis with satellite observations over East Asia from a radiance-based evaluation approach". In this way, the readers could easily distinguish this paper from the papers using conventional "retrieval-based approach".

**Response:** Thanks for the suggestion, and we have changed the title into "Evaluation of cloud properties from the reanalysis over East Asia with a radiance-based approach".

2. In the revised manuscript of line 276, the authors mentioned "To better illustrate the differences between cloudy and clear pixels, we distinguish them based on integrated column cloud optical depth in each pixel of 0.1". This kind of treating obvious will omit a large amount of cloudy pixels with low optical depth, especially the thin cirrus clouds. Could the authors estimate how this choice of cloud-clear pixel threshold will influence the results?

**Response:** Yes, if the cloudy and clear-sky pixels are generally distinguished based on the integrated COD of 0.1, some optically thin clouds may be missed. Figure 1 below shows the correlation coefficients between the observed and simulated BTs in the 11.2-μm as a function of different integrated column CODs. As the threshold decreases to 0.01 or smaller, the correlations between the observation and simulation achieve stable values. Deviations are obvious as the integrated column CODs increase from 0.01 to 0.1, especially for CRA. However, the general analysis is the same, and

there is no influence on our conclusion. Following the suggestion of the reviewer, we have updated Figure 5 in the revision by the following figure. The threshold between cloudy and clear-sky pixels is modified as 0.001, and the corresponding discussions are updated.

[Figure]

**Figure 1.** Correlation coefficient between observed and simulated BTs in the 11.2-μm with pixels for clear (green) and cloudy (red). The cloudy and clear-sky is distinguished by the threshold based on integrated column cloud optical depth (COD).

[Figure]

**Figure 2.** Comparisons between the observed and simulated BTs in the 11.2-μm channel with pixels for cloudy (top) and clear-sky (bottom). The distinction between cloudy and clear-sky is based on integrated column cloud optical depth of 0.001 and the results are taken at 00:00 (UTC) on 12 September 2016.

3. This manuscript still has a lot of typos and errors, and thus I suggest the authors thoroughly check their paper for the small typos. For example, in the abstract, quite a few acronyms are actually not needed to be defined at all, because they are never used again, i.e., RTM, AHI, BTDs. And in the main text, many acronyms are defined too many times (more than once), such as AHI, again. And in line 349, "clear seasonable variation" – I think the authors mean "clear seasonal variation"? Mostly, such typos are small errors, but they just make the paper look bad. I hope the authors could take the chance to polish the paper before it is finally accepted for publication.

**Response:** Thanks for the suggestions and comments. We have thoroughly and carefully checked the manuscript. The typos and errors are corrected, and some sentences and paragraphs are rephrased.